

# Snow thermal conductivity controls future winter carbon emissions in shrub-tundra

Johnny Rutherford[1], Nick Rutter[1], Leanne Wake[1], Alex J. Cannon[2]

[1]Department of Geography and Environmental Sciences, Northumbria University, Newcastle Upon Tyne, UK.
[2]Climate Research Division, Environment and Climate Change Canada.

*Correspondence to*: Johnny A. D. Rutherford (Jonathan.a.d.rutherford@northumbria.ac.uk)

**Abstract**

The Arctic winter is disproportionately vulnerable to climate warming and approximately 1700 Gt of carbon stored in high

latitude permafrost ecosystems is at risk of degradation in the future due to enhanced microbial activity. Few studies have

been directed at high-latitude cold season land-atmosphere processes and it is suggested that the contribution of winter

season greenhouse gas (GHG) fluxes to the annual carbon budget may have been underestimated. Snow, acting as a thermal

blanket, influences Arctic soil temperatures during winter and parameters such as snow effective thermal conductivity ($K_{eff}$)

are not well constrained in land surface models which impacts our ability to accurately simulate wintertime soil carbon

emissions. A point-model version of the Community Land Model (CLM5.0) forced by an ensemble of NA-CORDEX (North

American Coordinated Regional Downscaling Experiment) future climate realisations (RCP 4.5 and 8.5) indicates that

median winter $CO_2$ emissions will have more than tripled by the end of the century (2066-2096) under RCP 8.5 and using a

$K_{eff}$ parameterisation which is more representative of Arctic snowpack.  Implementing this $K_{eff}$ parameterisation increases

simulated winter $CO_2$ in the latter half of the century (2066-2096) by 130% and $CH_4$ flux by 50% under RCP 8.5 compared

to the widely used default $K_{eff}$ parameterisation. The influence of snow $K_{eff}$ parameterisation within CLM5.0 on future

simulated $CO_2$ and $CH_4$ is at least as significant, if not more so, than climate variability from a range of NA-CORDEX

projections to 2100. Furthermore, CLM5.0 simulations show that enhanced future air and soil temperatures increases the

duration of the early winter (Sept-Oct) zero-curtain, a crucial period of soil carbon emissions, by up to a month and recent

increases in both zero-curtain and winter $CO_2$ emissions appear set to continue to 2100. Modelled winter soil temperatures

and carbon emissions demonstrate the importance of climate mitigation in preventing a significant increase in winter carbon

emissions from the Arctic in the future.

## 1 Introduction

It is estimated that ~1700 Gt of carbon is stored in permafrost ecosystems of northern latitudes (Miner et al., 2022), which

accounts for half of the global soil carbon storage (Hugelius et al., 2014), and is at risk of degradation due to climate change

(Koven et al., 2013). Near surface permafrost temperature has been increasing 1.1°C per decade since 1987 (AMAP, 2017)

which acts to reduce near-surface permafrost extent and increase active layer thickness (Intergovernmental Panel on Climate,



2023); such loss of permafrost drives further release of carbon in the form of $CO_2$ during winter. Arctic winters are estimated to warm by $4.8^\circ C$ by 2100 compared to $2.2^\circ C$ in the summer over the same period (Webb et al., 2016, Christensen et al., 2013). Climate warming has enhanced the warming of soil at Arctic sites in the recent past (AMAP, 2017, Ednie and Smith, 2015) and future warming will enhance microbial decomposition of soil organic matter, driving subsequent release of carbon dioxide ($CO_2$) and methane ($CH_4$) (Natali et al., 2019, Schuur et al., 2015). Furthermore, the winter carbon flux is crucial to the annual carbon budget, as demonstrated through flux measurements from a tundra site, where approximately 60% of $CO_2$ uptake in the growing season was then lost in the non-growing period, and $CH_4$ emissions during this period accounted for 30% of the annual budget (Kittler et al., 2017). Few studies have been directed at cold season land-atmosphere processes and it is suggested that the contribution of winter season greenhouse gas (GHG) fluxes to the annual carbon budget may have been underestimated (Pongracz et al., 2021). Future winter emissions will likely offset growing season uptake under Representative Concentration Pathways (RCP) 4.5 (medium global emission scenario) and 8.5 (high global emission scenario) (Natali et al., 2019). Additionally, the Arctic winter is longer than the growing season (the latter being typically 2-5 months in length), and therefore may represent a substantial input of carbon to the atmosphere that is not accurately represented in current climate models nor quantified accurately in global carbon budgets (Treat et al., 2024). Assessing simulations of heat and gas fluxes to and from subnivean soils is therefore critical in understanding how well models, such as The Community Land Model (CLM5.0; (Lawrence et al., 2019), can be expected to simulate future carbon cycling in the Arctic.

The hydrological cycle in the Arctic is projected to intensify throughout the 21$^{st}$ century and new model estimates from the Coupled Model Intercomparison Project 6 (CMIP6) support a future transition from a snow- to rain-dominated Arctic in the summer and autumn by 2100 (McCrystall et al., 2021). Despite future temperatures still being cold enough to enhance snowfall, winter rainfall also intensifies, with projected increases even matching that of snow (Bintanja and Andry, 2017). Snow is a key determinant of ground temperature and-freeze thaw state through its insulating properties (Bigalke and Walsh, 2022). The prospect of a rain dominated Arctic has implications for soil temperature, through changes in snow cover duration (Mudryk et al., 2019, Mudryk et al., 2020), insulative and structural snow properties (Cohen et al., 2015, Serreze et al., 2021), and increased soil moisture (Trenberth, 1998, Cohen et al., 2015). In the recent past (1979-2018) snowpack properties feature significant increases in spring snow bulk density (May and June), a downward trend in snow cover duration and an upward trend in wet snow, all of which result in a soil surface temperature increase of +0.41 K decade$^{-1}$ in winter (Royer et al., 2021). Such temperature increases, induced by snowpack evolution may contribute to increased soil heterotrophic respiration and carbon release ($CO_2$ and $CH_4$) in future winters (Natali et al., 2019). Model estimates, driven by environmental metrics from CMIP5, indicate by 2100 Arctic winter $CO_2$ emissions will have increased by 17% under RCP 4.5 and by 41% under RCP 8.5 (Natali et al., 2019). Winter $CH_4$ emissions from high latitude wetlands (>60°N) were previously assumed to be negligible (Treat et al., 2018) but recent observations show that they can be substantial sources of carbon in the winter (Ito et al., 2023). A more recent model intercomparison has shown that nearly one third of annual $CH_4$ emissions occurred during Sept-May and new observations suggest it may be greater than one third (Ito et al., 2023).



The 'shoulder' seasons, defined here as the early (after summer plant senescence has ended) and late (before snowmelt is complete) winter period (Natali et al., 2019), are a critical period in the context of the annual Arctic carbon budget, especially in terms of $CH_4$ where zero-curtain (when soil temperatures are poised near $0^{\circ}C$, such as in early winter) emissions alone

contribute around ~20% of the annual budget (Zona et al., 2016). In terms of $CO_2$, historic early winter season emission rates have increased $73\% \pm 11\%$ since 1975 (October-December) (Commane et al., 2017) in response to a warming climate, demonstrating the vulnerability of this period to future warming. To date Earth System Models (ESMs) have poorly simulated respiration in the early winter shoulder season (Commane et al., 2017) which is a key component of the annual Arctic carbon budget (Kittler et al., 2017).


Land Surface Models (LSMs) such as CLM5.0 struggle to accurately reproduce cold-season $CH_4$ and $CO_2$ emissions due to uncertain numerical representation of mechanisms controlling heterotrophic respiration (Zona et al., 2016, Commane et al., 2017, Tao et al., 2021, Natali et al., 2019). One such mechanism is the mediation of winter soil temperatures by the insulating effect of seasonal snow. The default parameterisation of snow thermal conductivity ($K_{eff}$, W $m^{-1}$ $K^{-1}$, Jordan (1991)) in CLM5.0

produces winter soil temperatures that are colder than observations (Dutch et al., 2023) which may contribute to an underestimation of soil respiration in the early winter (Commane et al., 2017). Sturm et al. (1997) presented an alternative parameterisation for $K_{eff}$ derived from extensive Arctic snow measurements (n=488) that was implemented into CLM5.0 by Dutch et al. (2023). The Sturm parameterisation is more relevant for Arctic snow types compared to Jordan (1991), as it accounts for snow layers such as basal depth hoar and wind-slab within the snowpack and produced more accurate simulations

of subnivean soil temperature (Dutch et al., 2023). Further, CLM5.0's current default values of soil moisture threshold for decomposition ($\Psi_{min}$) and respiratory response to changes in temperature (Q10 and Q10ch4) are also not appropriate for Arctic tundra environments (Dutch et al., 2023) and, alongside $K_{eff}$, require adjustment when running CLM5.0 simulations of soil respiration (SR) and methane flux (FCH4) in these settings.

This study assesses the sensitivity of SR (i.e. soil $CO_2$ flux to the atmosphere during winter) and $CH_4$ flux to $K_{eff}$, $\Psi_{min}$, Q10 and Q10ch4 within CLM5.0, and presents future projections of carbon emissions for Trail Valley Creek to 2100. Firstly, future changes in Arctic meteorology at TVC are analysed using an ensemble (n=33) of future NA-CORDEX projections (McGinnis and Mearns, 2021), representing varying Global Climate Model-Regional Climate Model (GCM-RCM) combinations; RCP scenarios (4.5; n=6 and 8.5; n=27) and RCM resolutions (0.22 degrees; n=11, and 0.44 degrees; n=22). The NA-CORDEX

ensemble is then applied to a point model of the default version CLM5.0 ('CORDEX-Jordan') to simulate present (2016-2046) and future (2066-2096) Snow Water Equivalent (SWE), 10cm Ground Temperature (GT10) and Soil Moisture (SOILLIQ) and resulting winter SR and $CH_4$ fluxes. The ensemble of experiments is then re-run, with the representation of $K_{eff}$ changed from Jordan *et al.* (1991) to Sturm *et al.* (1997) to assess the impact on soil temperature; this ensemble is named 'CORDEX-Sturm'. Finally, a plausible parameter space of $\Psi_{min}$, Q10 and Q10ch4 for Arctic environments is applied to each experiment grouping



(CORDEX-Jordan and CORDEX-Sturm) to assess the effect of assumptions governing the moisture- and temperature-related controls on soil respiration, resulting in an overall set of 396 experiments to quantify the effects of parametric and future climate uncertainty on carbon emissions at TVC.

## 2 Methodology

### 2.1 Study Location

Point scale CLM5.0 simulations are produced for Trail Valley Creek (TVC; 68° 45' N, 133° 30' W), a research basin on the boundary of the boreal-tundra ecotone ~50km north of Inuvik in the Inuvialuit Settlement Region of western Inuit Nunangat, in the Western Canadian Arctic. The basin lies within the Inuvialuit Settlement Region near the Inuvik-Tuktoyaktuk Highway. Vegetation consists of shrub tundra and is dominated by mineral earth hummocks. Mean annual air temperature from 1999-2018 averaged -7.9°C (Grünberg et al., 2019) with average seasonal variations of +10°C in summer and -25°C in winter (1995)

(Marsh et al., 2002). Air temperatures increased 1.1°C per decade during the 1990-2018 period with the strongest warming observed in May (2.8°C per decade) (Grünberg et al., 2019). Snow depth at TVC is typically <50cm, with deeper packs associated with variations in topography and vegetation (King et al., 2018). Snow is often re-distributed by wind resulting in scour and snow drift features (Thompson et al., 2016). Snow cover typically lasts for 8 months (Oct-May) and underlying permafrost is between 350-500m in thickness with an active layer of 0.5-1.0m (Wilcox et al., 2019, Dutch et al., 2023). Tundra

environments occupy ~31% of the Canadian Arctic (Bliss and Matveyeva, 1992, Quinton et al., 2000) and well-drained, shrub- and lichen covered uplands, such as those at TVC, comprise 80% of the Arctic-boreal region (Voigt et al., 2023). TVC has long lasting concurrent hydrometeorological and Eddy Covariance (EC) carbon flux data since 2013 (Tutton, 2024) which can be used to bias correct future meteorological simulations (see section 2.2).

### 2.2 Meteorological forcing data and simulations using CLM5.0

NA-CORDEX meteorological forcing data (incoming shortwave and longwave radiation, precipitation, humidity, wind speed, air pressure, min and max temperature) for the grid cell closest to TVC provides CLM5.0 with daily meteorological conditions to 2100. This forcing dataset was generated from a suite of 33 bias-corrected GCM-RCM combinations for North America (McGinnis and Mearns, 2021) ran under full transient conditions with a historical period spanning 1950-2005 and with scenarios RCPs 4.5 and 8.5 for 2006-2100 at 0.22 and 0.44 degree resolution. To minimise the influence of climate model

bias on model results, meteorological forcing data from the climate models were bias-adjusted to match statistical characteristics of meteorological observations (Tutton, 2024) in the overlapping 1992-2022 period following Cannon (2018) and Cannon et al. (2022). CLM5.0 was run in "point mode" (a 0.1° x 0.1° grid cell) and centred at the location of TVC as per Dutch et al. (2021). Model spin up is required to stabilise carbon pools within CLM5.0 prior to model initiation; model simulations were spun-up for 512 years and full details of these spin-up conditions are outlined in Dutch et al. (2023).





## 2.3 CLM5.0 parameter assessment

$K_{eff}$ describes the rate of heat transfer from the atmosphere through the snow to underlying soils and as such $K_{eff}$ influences heterotrophic respiration rates and subsequent production of $CO_2$ and $CH_4$. CLM5.0 default $K_{eff}$ parameterisation using Jordan *et al.* 1991 overestimates $K_{eff}$ by a factor of 3 for tundra snow (Dutch et al., 2021). In CLM5.0, $K_{eff}$ is parameterised as a function of snow density (Kg m$^{-3}$) (equation 1), which is calculated from relative proportions of ice mass ($m_i$) and liquid water mass ($m_{lw}$), weighted by the snow cover fraction ($F_{sno}$) for each grid cell. $K_{eff}$ can then by calculated using coefficients in equation 2 (Jordan *et al.* 1991), or equation 3 (Sturm *et al.* 1997):

$$\rho = \frac{m_i \times m_{lw}}{F_{sno} \times h_{sl}} \qquad (1)$$

$$K_{eff} = K_{air} + (((7.75 \times 10^{-5} \times \rho) + (1.105 \times 10^{-6} \times \rho^2))(K_{ice} - K_{air})) \qquad (2)$$

$$\begin{cases} K_{eff} = 0.138 - 1.01\rho + 3.233\rho^2, & 0.156 \le \rho \le 0.6 \\ K_{eff} = 0.023 + 0.234\rho, & \rho < 0.156 \end{cases} \qquad (3)$$

The use of the snow thermal conductivity parameterisation following Sturm et al. (1997) improved soil temperature simulations in CLM5.0 at TVC (Dutch et al., 2021) and at other sites with additional land surface models (Royer et al., 2021). Here, the Sturm et al. (1997) $K_{eff}$ parameterisation was applied to future CLM5.0 simulations and compared with the default parameterisation which follows Jordan (1991). CLM5.0 uses $\Psi_{min}$ to describe the soil moisture threshold for respiration (MPa). Without adjustments to $\Psi_{min}$, CLM5.0 simulates near-zero soil respiration for the majority of the snow-covered non-growing season which is not consistent with field observations (Dutch et al., 2023). Reducing $\Psi_{min}$ facilitates respiration in moisture-limited soils (Tao et al., 2021). $\Psi_{min}$ is applied in CLM5.0 as described by Andren and Paustian (1987):

$$r_W = \sum_{j=1}^{5} \begin{cases} 0 & \text{for } \Psi_j < \Psi_{min} \\ \frac{log(\Psi_{min}/\Psi_j)}{log(\Psi_{min}/\Psi_{max})} w_{soil,j} & \text{for } \Psi_{min} < \Psi_j < \Psi_{max} \\ 1 & \text{for } \Psi_j > \Psi_{max} \end{cases} \qquad (4)$$

where $\Psi_j$ is the soil water potential in soil layer $j$, and $\Psi_{min}/\Psi_{max}$ are upper and lower limits. The default values of $\Psi_{min}/\Psi_{max}$ for CLM5.0 are -2 MPa and -0.002 MPa respectively.

As well as $K_{eff}$ and $\Psi_{min}$, we also focus on the temperature soil decomposition modifier $r_T$ which is parameterised using a Q10 function which describes the dependence of biological metabolic processes, such as respiration, on temperature. $r_T$ is calculated as follows:



$r_T = Q_{10}^{(\frac{T_j - T_{ref}}{10})}$           (5)

Where $T_j$ is the temperature of soil layer $j$ and $T_{ref}$ is a reference temperature with a default value of 25°C. CLM5.0 uses a fixed Q10 of 1.5, however, this is unrealistic for Arctic ecosystems which typically have an average Q10 of 5.4 (Chen et al., 2020). A range of $\Psi_{min}$ ('minpsi_hr', -2 to -20) and Q10 ('q10_hr', 'froz_q10', 1.5 to 7.5) was used to assess the sensitivity of

CLM5.0 for future simulations to 2100. A parameter 'q10ch4' within the methane module of CLM5.0 which controls temperature dependency of methane production is available and was tested from its default value 1.33 to a plausible extreme of 4 (Müller et al., 2015, Riley et al., 2011) with a reference temperature of 22°C. The full range of tested parameters are presented in Appendix C.

A major focus of this study is the winter season when snow is on the ground (i.e. the snow-covered non-growing season). We

define this period as the time when all model ensemble members (RCP 4.5 n=6, RCP 8.5 n=27) agree that SWE is >5mm. Simulations of SR and FCH4 are filtered by these constraints so the analysis in Figures 5 and 6 is focused only on carbon fluxes across a common snow-covered, non-growing season in all scenarios and forcing datasets. The resulting data is therefore weighted towards a shorter snow-cover period as ensemble members capture a range of snow-cover durations. Periods of the winter season that are critical to driving the timing and rates of respiration are described as follows. The 'zero-curtain period'

is defined as days where soil temperature falls between -0.75 and 0.75°C (Tao et al., 2021, Zona et al., 2016) while 'shoulder seasons' describes annual transitional periods of snow melt (Apr-May) and plant senescence-to-freeze up (Sep-Oct) (Shogren et al., 2020). Testing for statistically significant differences between distributions of future meteorological conditions and median winter carbon fluxes are evaluated using the non-parametric Kolmogorov–Smirnov (K-S test) test.





## 3 Results

**3.1 NA-CORDEX meteorological forcing data**

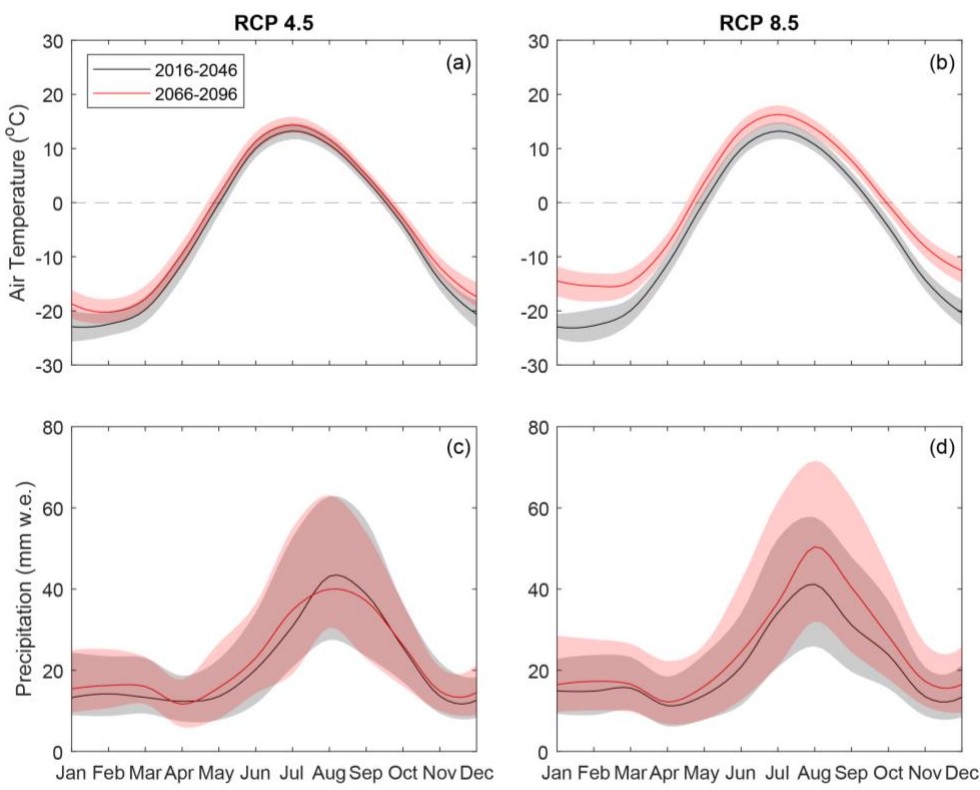

**Figure 1** – Simulated median monthly precipitation and temperature over two 30-year time periods: 2016-2046 (black) and 2066-2096 (red) under RCP 4.5 and RCP 8.5, for TVC using meteorological data from an ensemble of 33 NA-CORDEX GCM-RCM combinations (RCP 4.5 n=6, RCP 8.5 n=27). Shaded areas represent the 25th and 75th percentiles (variability between individual ensemble members for precipitation is presented in Appendix A).

Bias-corrected NA-CORDEX model ensemble projections show an increase in future winter air temperatures at TVC, which intensifies under RCP 8.5, with a 9°C increase from present to future in median January air temperatures (Figure 1a and b). The timing and rate of air temperature increase in the spring is comparable between present and future, whereas the rate of cooling after the summer maximum decreases in the future under RCP 8.5 (a decrease of 6.8°C/month in the present and 5.7°C

/month in the future for Jul-Dec). Under RCP 8.5, significant increases in precipitation are projected from Jul-Dec, the largest increase occurring in August from 41 to 50 mm w.e. (Two sample K-S test, test statistic: D=0.16, p<0.05, Figure 1d), Warmer future air temperature influences the phase of future precipitation with more precipitation falling as rain than snow, especially in transitional shoulder seasons between summer and winter (Figure 2).



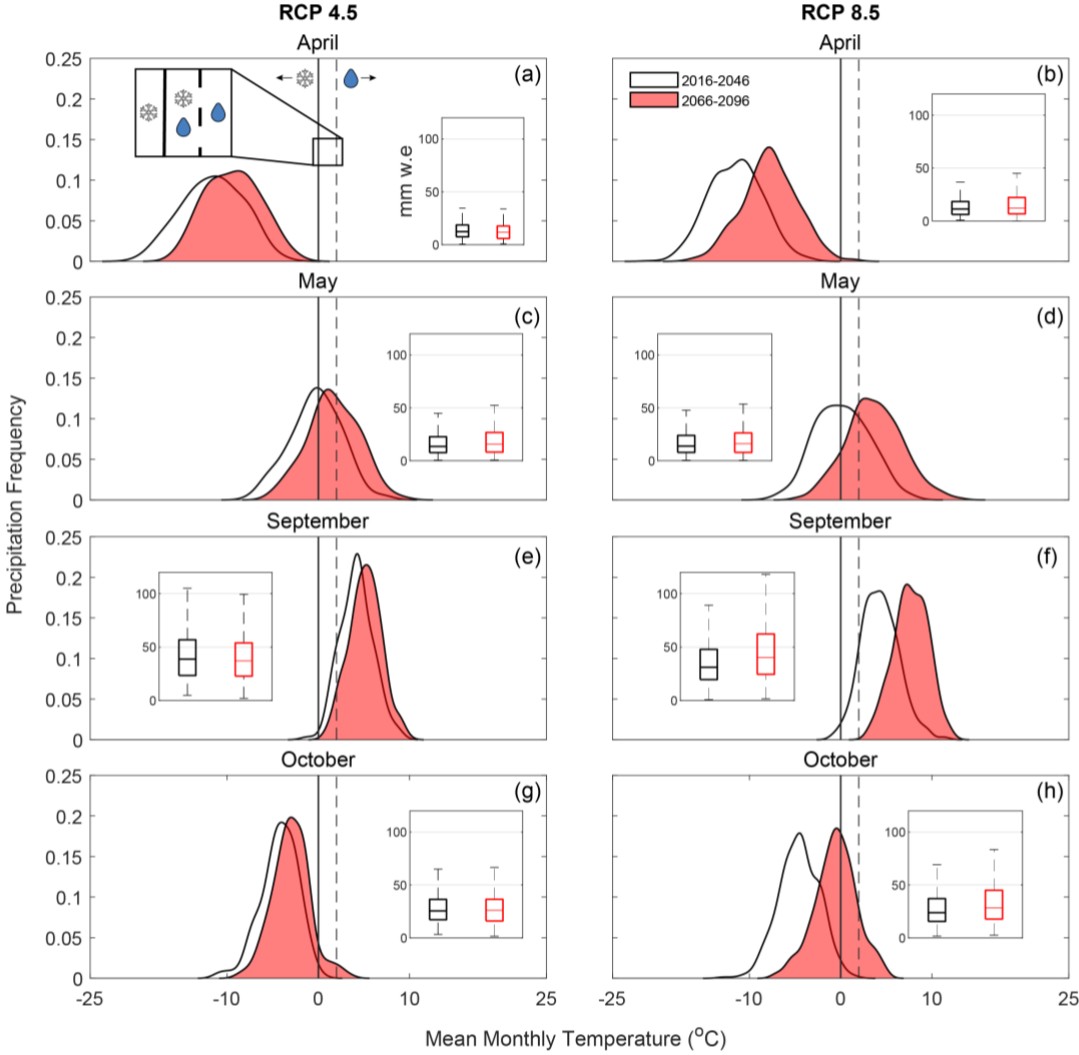

**Figure 2 –** Simulated median monthly precipitation vs mean monthly temperature over two 30-year time periods: 2016-2046 (black) and 2066-2096 (red) under RCP 4.5 and RCP 8.5, for TVC using meteorological data from an ensemble of 33 NA-CORDEX GCM-GCM combinations (RCP 4.5 n=6, RCP 8.5 n=27). Inset boxplots show monthly total precipitation for the two 30-year periods.

Changes in air temperature in May and October cause a phase shift (snow to rain) in total monthly precipitation under RCP 8.5 from present to future (Figure 2, d and h). In May, in the present (2016-2046) 50% of precipitation events in the current period fall within 0-20°C (as rain, or as a mix of snow and rain) compared to 87% in the future (2066-2096). In October, only 2% of precipitation events fall within 0-20°C compared to 40% in the future. Furthermore, in May, future (red) peak precipitation frequency occurs at 2.7°C compared to -0.6°C at present (black), and in October at -0.5°C compared to -4.6°C at present (Figure 2) under RCP 8.5 which indicates a shoulder season precipitation shift from majority snow to a mix of rain/snow. By contrast, differences between current and future total precipitation are small (Figure 2, a-h, insets), and increases





from present to future are significant only in September (Two sample K-S test: 31 to 40mm, D=0.17, p<0.05) and October (Two sample K-S test: 23 to 28mm, D=0.12, p<0.05) under RCP 8.5. As such, changes in magnitude and phase of precipitation in the shoulder seasons may influence future snow-season length, snow thermal conductivity (due to an increase in melt conditions) and subsequent temperature and moisture of soils.

## 3.2 CLM5.0 simulations

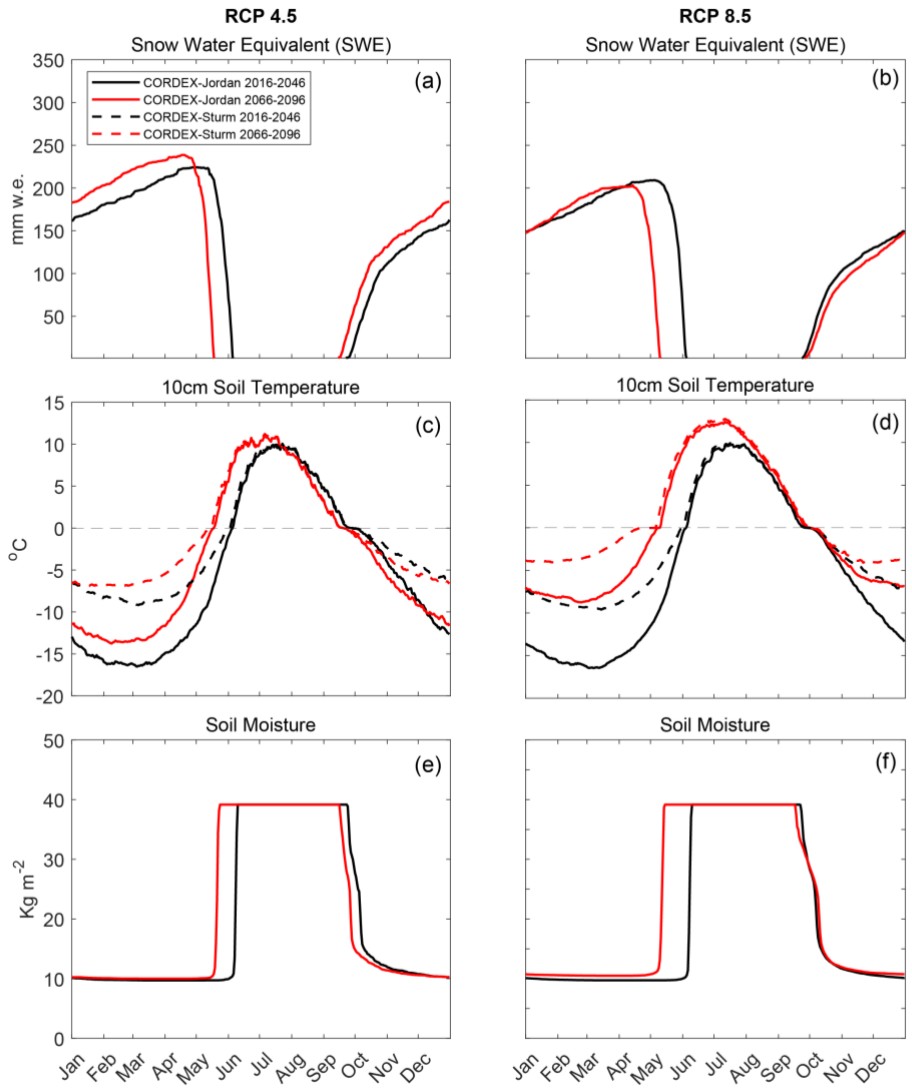

**Figure 3** – CLM5.0 simulated median daily snow water equivalent (SWE; a,b), 10cm soil temperature (c,d) and soil liquid water (12cm) content (e,f) over two 30-year time periods: 2016-2046 (black) and 2066-2096 (red) under RCP 4.5 and RCP 8.5, for TVC using input meteorological data from an ensemble of 33 RCM-GCM combinations (RCP 4.5 n=6, RCP 8.5 n=27). Solid and dashed lines show ensemble median values for CORDEX-Jordan and CORDEX-Sturm experiments respectively.





An increase in future rainfall in September impacts both the timing of winter onset of snow accumulation and the magnitude of peak SWE later in the winter (Figure 3). The duration for winter snow cover is shorter in the future where because future spring melt-out occurs between 18 to 25 days earlier in both RCP scenarios, while the timing of initial snow accumulation remains similar (+/- 4-6 days). Projected future peak annual SWE shows an increase of 14 mm w.e. under RCP 4.5 and a decrease of 6 mm w.e. under RCP 8.5 and early spring snow melt is coincident with earlier increases in soil temperature in

mid-late spring (Figure 3, a-d).

Using CORDEX-Sturm, the median minimum winter soil temperature is 4-7°C warmer by the end of the century (2066-2096) compared to CORDEX-Jordan under both RCP scenarios (**Error! Reference source not found.**,c,d). Warmer air temperatures predicted for the end of this century (Figure 1, a,b) will warm the soil, and snow cover will then insulate the soil from the colder atmosphere, maintaining these elevated soil temperatures throughout the winter (Figure 3,c,d).

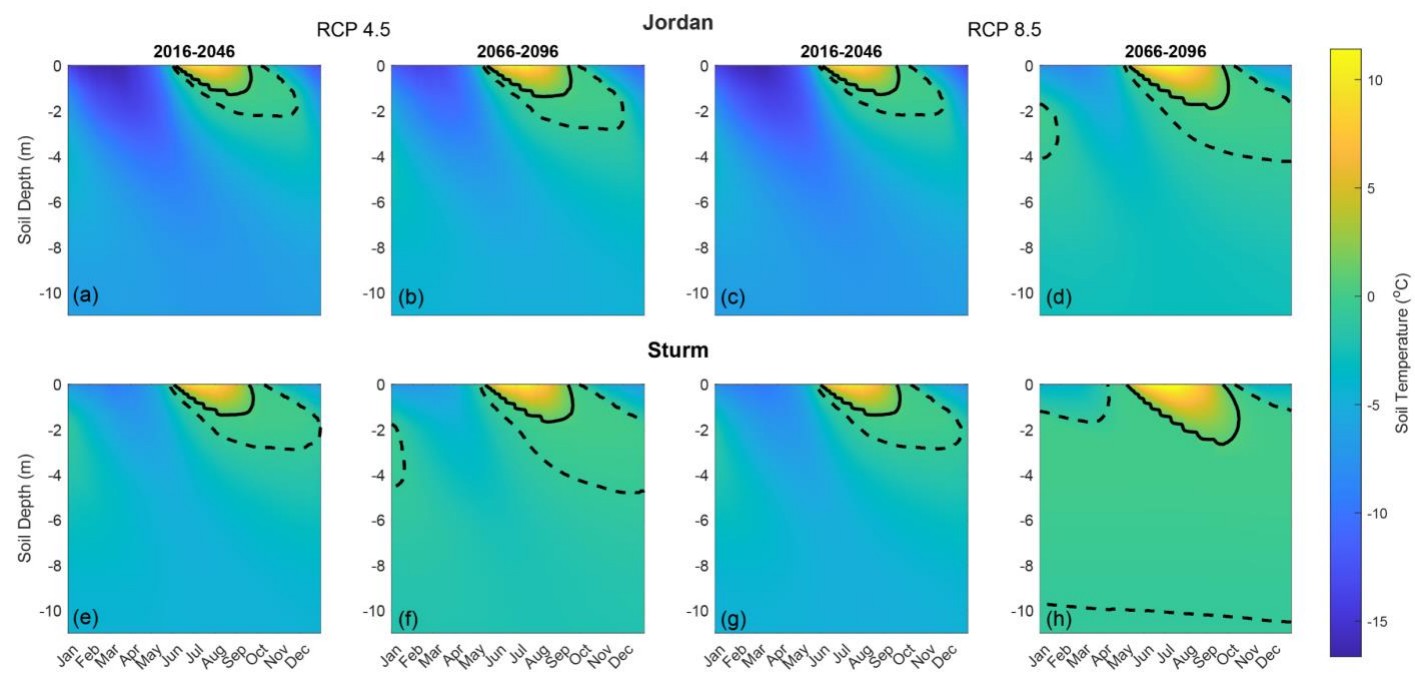


**Figure 4 –** CLM5.0 simulated median daily soil temperature with depth over two 30-year time periods: 2016-2046 and 2066-2096 under RCP 4.5 (left) and 8.5 (right). 1st row plots show CLM5.0 under default K$_{eff}$ parameters (Jordan et al. 1991) and 2nd row plots show an alternative parameterization for snow thermal conductivity (Sturm et al. 1997). Solid black lines indicate 0.75°C and dashed -0.75°C, indicating zero curtain between them.

Due to earlier snow melt, soils are predicted to thaw earlier in the year under both RCP scenarios (RCP 4.5 = 19 days, RCP 8.5 = 26 days; Figure 3, e, f). The timing of soil moisture increase in May and June remains similar in the present (2016-2046) between RCP 4.5 and 8.5 but occurs 20-30 days earlier in the future (2046-2096). By comparison, the timing of soil moisture decrease in September and October remain similar at Julian day 257-267 between the two RCP scenarios and 30-year time

 

periods. Modelled soil moisture plateaus at 10 kg m$^{-2}$ from October-May and reaches saturation 39.167 kg m$^{-2}$ from late May-

early October.

Evidence of a late winter (February-March) zero-curtain period is minimal across all but one scenario (Figure 4h), however soil temperatures at depths up to 10m suggest that the zero-curtain duration from Aug-Dec increases by 10 and 36 days in the future under CORDEX-Jordan and CORDEX-Sturm respectively (Appendix B). In the future under RCP 8.5, early winter near-zero temperatures penetrate up to 6m deeper into the soil column compared with the present day in the CORDEX-Sturm

simulations (Figure 4h). Such an increase demonstrates both the influence of snow on soil temperatures at depth and the risks of climate warming on permafrost degradation and possible mobilisation of legacy carbon from Arctic soils. An earlier spring snow melt, increasing winter soil temperatures, and longer zero-curtain duration combine to create the conditions for increased cumulative seasonal carbon emissions in the future at TVC.

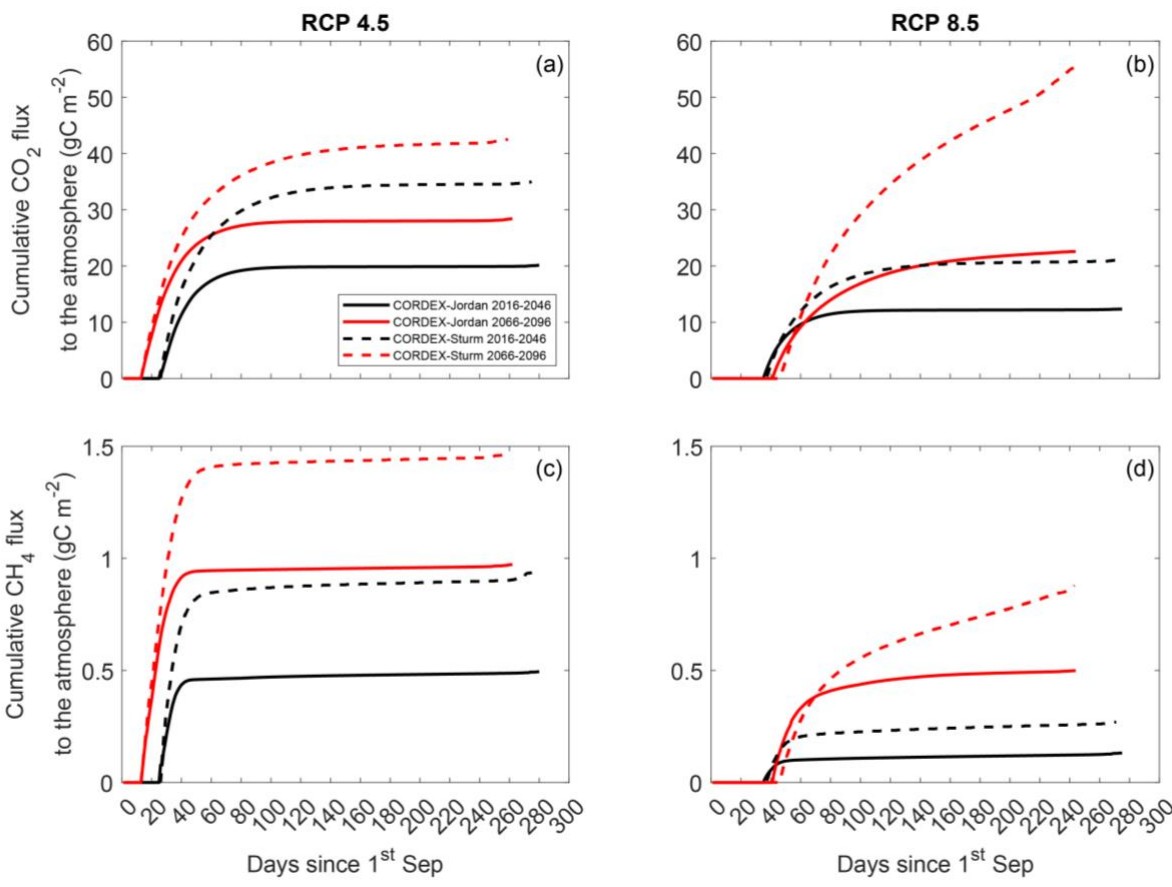

**Figure 5** – CLM5.0 simulated cumulative CO$_2$ and CH$_4$ flux to the atmosphere during snow-cover periods over two 30-year time periods: 2016-2046 (black) and 2066-2096 (red) under RCP 4.5 and RCP 8.5, for TVC using meteorological forcing from an ensemble of 33 RCM-GCM combinations (RC P 4.5 n=6, RCP 8.5 n=27). Solid lines show median values from CORDEX-Jordan ensemble and dashed lines show values for the CORDEX-Sturm ensemble. Variable plot line lengths are indicative of changes in snow-season length.





Cumulative snow-covered $CO_2$ and $CH_4$ fluxes are projected to increase under both RCP scenarios (Figure 5) with the highest
$CO_2$ increase produced for CORDEX-Sturm (34.2 gC m$^{-2}$; from 21.1 to 55.3 gC m$^{-2}$) under RCP 8.5. A later onset of snow in
2066-2096 facilitates earlier emissions of $CO_2$ and $CH_4$ during the Autumn under RCP 4.5 (12-13 days post Sept 1st; Figure 5,
a, c) compared with RCP 8.5 (41-47 days post Sept 1st; Figure 5, b, d). Delayed snow onset reduces overall cumulative carbon
emissions during the snow-cover period under RCP 8.5 (Figure 5, b, d) compared with RCP 4.5 (Figure 5, a, c) which suggests
that the contribution of early winter emissions to the overall accumulation of winter emissions is key, in the period where the
snowpack is accumulating and soil temperatures are cooling. The importance of the early winter $CO_2$ release to the overall
carbon budget under RCP 4.5 is clear as there is minimal increase in cumulative carbon after mid-late November with increases
of 0.02 gCO$_2$ m$^{-2}$ day$^{-1}$ and 0.00015 gCH$_4$ m$^{-2}$ day$^{-1}$ (day 80 to day 240) averaged across present/future and CORDEX-
Jordan/Sturm. By contrast, under RCP 8.5, $CO_2$ emissions are prolonged into the winter due to warmer soil temperatures, with
an average increase of 0.07 gCO$_2$ m$^{-2}$ day$^{-1}$ and 0.003 gCH$_4$ m$^{-2}$ day$^{-1}$ (day 80 to day 240) (Figure 5). Applying Sturm's snow
thermal conductivity scheme within CLM5.0 increases the total accumulation of winter $CO_2$ at TVC by 49-150% and $CH_4$ by
50-74% across both RCP scenarios compared to CORDEX-Jordan, underlining the importance of snow representation in
modelling cumulative soil carbon emissions. Appendix D presents a comparison between emissions during the snow-covered
season and those unconstrained by snow cover. The cumulative emissions not limited by the snow-covered period are 3-4
times higher than those constrained by snow and highlight the impact of the early winter period on accumulated winter soil
carbon emissions.

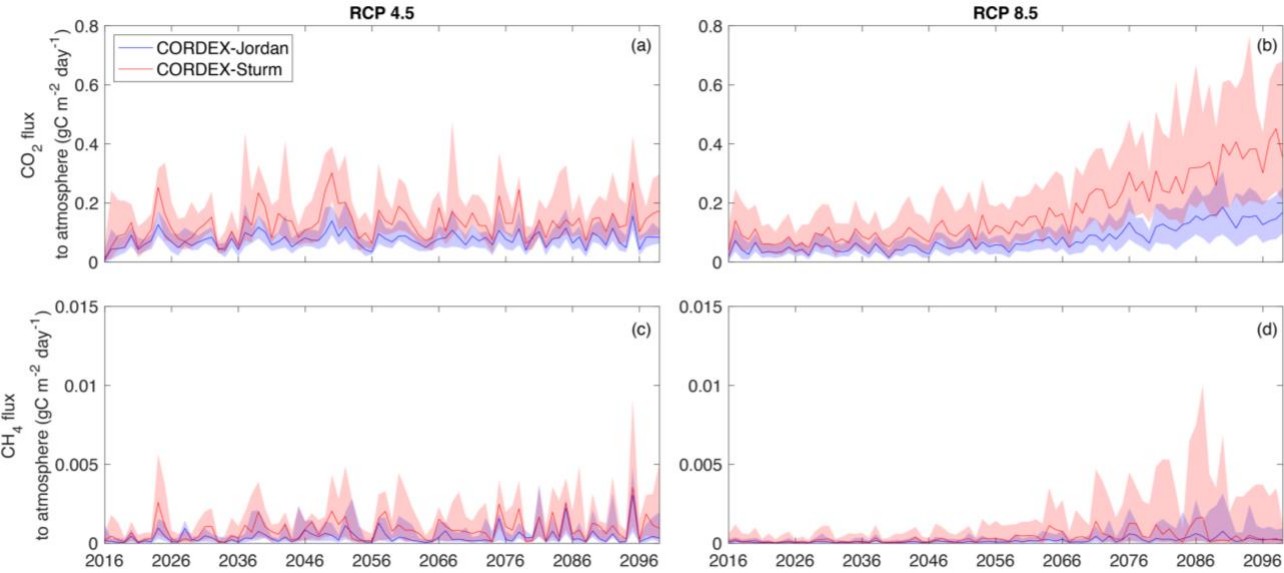

**Figure 6** – CLM5.0 simulated median soil respiration for the winter season, comparing the CORDEX-Jordan ensemble with CORDEX-Sturm for two RCP scenarios (RCP 4.5 n=6, RCP 8.5 n=27). Shaded areas represent 75th and 25th percentiles and represent the ensemble distribution. Ensemble median and uncertainty calculations incorporate a range of $\Psi_{min}$, Q10 and Q10ch4 (Appendix E).



From 2016-2046 to 2066-2096, median winter $CO_2$ flux increases by 18% (0.11 – 0.13 $gCO_2$ m$^{-2}$ day$^{-1}$) under RCP 4.5 and more than triples (0.08 to 0.3 $gCO_2$ m$^{-2}$ day$^{-1}$) under RCP 8.5, using CORDEX-Sturm. CORDEX-Sturm significantly increases winter $CO_2$ emissions compared with CORDEX-Jordan, by 57% (2016-2046) to 62% (2066-2096) in RCP 4.5 and 100% (2016-2046) to 130% (2066-2096) in RCP 8.5 (Two sample K-S test: D=0.26, p<0.01, Figure 6). Likewise, under RCP 8.5 CORDEX-Sturm significantly increases median $CH_4$ emissions by 50% during 2066-2096 compared to CORDEX-Jordan

(2.5e-04 to 3.8e-04 $gCH_4$ m$^{-2}$ day$^{-1}$) (Two sample K-S test: D=0.13, p<0.01, Figure 6) further showing the impact of snow representation on the simulation of $CH_4$ from Arctic soils. Between RCP 4.5 and 8.5 median $CO_2$ emissions more than double (0.13 to 0.30 $gCO_2$ m$^{-2}$ day$^{-1}$) for the last 30 years of the century using CORDEX-Sturm. CLM5.0 simulations indicate that warmer soil conditions induced by climate warming under RCP 8.5 intensify winter carbon emissions from TVC soils in the future.

**4 Discussion**

**4.1 Future temperature, precipitation, and snow cover**

NA-CORDEX forecasts suggest rising future winter air temperatures (Figure 1) which may increase winter soil heterotrophic respiration and contribute significantly to winter carbon emissions offsetting growing season carbon uptake in the future (Natali et al., 2019, Commane et al., 2017). Mobilisation of "legacy" carbon (Pedron et al., 2023) within Arctic permafrost is of

concern in a warming climate (Schuur et al., 2015) and warmer winters will enhance future climate change through carbon output from decomposition of soil carbon. Understanding changes in future Arctic air temperature and meteorology is key in determining the influence of a warming world on Arctic winter carbon emissions and vice versa.

NA-CORDEX ensemble simulations demonstrate a shift from a snow- to rain-dominated Arctic in the future, reported elsewhere in the literature (McCrystall et al., 2021) which is intensified under RCP 8.5 (Figure 2). The shift in precipitation

phase in September and October will lead to a change in snow properties, particularly the effective conductivity of the snowpack. Additionally, melt conditions and rain-on-snow events in the initial stage of the snow cover season provide energy for melt through condensation of water vapour onto the snowpack (Harr, 1981, Mazurkiewicz et al., 2008). An increase in precipitation fraction falling as rain is therefore a likely driver of earlier snowmelt in CLM5.0 simulations (Figure 3). Furthermore, the combination of higher air temperatures and increased rainfall under RCP 8.5 will reduce the insulating

capacity of snow on the soil due to densification of the snowpack (Marshall et al., 1999) and increase $K_{eff}$ which cools soils. Cooler soils will likely reduce projected carbon emissions from Arctic soils during thaw and freeze up, with causal effects on winter and annual carbon budgets. Future changes in precipitation have key, yet uncertain, implications for the Arctic surface and soil energy balance with regard to snow cover depth, timing and duration which are major factors controlling temperature variability in the upper 3 m soil (Callaghan et al., 2011).





## 4.2 Future soil carbon emissions

By the end of the century (2066-2096), CLM5.0 simulated median winter soil $CO_2$ emissions under RCP 4.5 are predicted to be less than half of those under RCP 8.5, showing that projected winter carbon emissions are tied to magnitude of global temperature increase. Winter snow cover has a significant influence on the ground thermal regime (Zhang, 2005) and it's representation within ESMs is critical for simulated soil temperatures. Under RCP 8.5 cumulative winter carbon emissions from soils at the TVC site are projected to increase in the future despite a reduction in snow-cover duration. The more realistic $K_{eff}$ parameterisation in the CORDEX-Sturm simulations results in reduced soil temperature biases in CLM5.0 (Dutch et al., 2023) and elevates simulated winter soil temperatures to between -10 and 0°C. Within this range, respiration rates begin to increase rapidly (Natali et al., 2019). Cumulative winter $CO_2$ simulations (Figure 5) under present day conditions are in line with contemporary (2016-2019) NEE simulations, generated using in-situ meteorological data at TVC Dutch et al. (2023) for both Jordan (~15-35 g$CO_2$ m$^{-2}$) and Sturm (~25-55 g$CO_2$ m$^{-2}$) parameterisations. Natali et al. (2019) estimated that winter $CO_2$ emissions would increase by 41% by 2100 (from 'present', i.e. 2003-2017 conditions; Natali et al. (2019); this study 2016-2046) under RCP 8.5 whereas CLM5.0 simulations of $CO_2$ emissions under CORDEX-Sturm more than triple under RCP 8.5 which demonstrates the impact of snow representation on simulated soil carbon emissions. Under RCP 8.5 the magnitude of the influence of CORDEX-Sturm on winter carbon fluxes is comparable to the uncertainty in the future climate, reinforcing the importance of snow representation in future projections of Arctic carbon fluxes. The response of carbon fluxes from permafrost zones is highly sensitive to hydrological change and an increase in $CO_2$ emissions post-2050 (Figure 6) is indicative of soil drying where, as temperatures increases, soil $CH_4$ production is strongly suppressed (Lawrence et al., 2015). Further, the difference in $CO_2$ output between RCP 4.5 and 8.5 shows the possible impacts of climate mitigation efforts on future Arctic winter carbon emissions (Figure 6).

The zero-curtain is an important period for facilitating cold season emissions from tundra ecosystems (Tao et al., 2021) and an increasing proportion of the soil at or around 0°C presents a risk of elevated carbon emissions from Arctic soils in the future. Persistent carbon emissions throughout the winter period are partly attributable to a permanently unfrozen active soil layer (Zona et al., 2016) and its influence in the future will be intensified by a longer zero-curtain period. Simulations performed in this study indicate that the maximum duration of early winter zero-curtain is projected to lengthen by as much as a month in the future, with soil temperatures at or near $0°$C also extending to depths up to 6 meters deeper than current conditions (Figure 4). Increases in the duration of the zero-curtain is concurrent with measured borehole data from 2006-2015 at an Arctic tundra site, which show an increase of up to 20 days, with large emissions of $CO_2$ between September and December in years with a longer zero-curtain period (Euskirchen et al., 2017, Larson et al., 2021). A deeper active layer (Aalto et al., 2018) and an increase in unfrozen soil (Schaefer and Jafarov, 2015, Natali et al., 2019, Elberling and Brandt, 2003) as a result of a longer and deeper zero-curtain will increase soil respiration (SR) in the future. In the recent past (1950 to 2017), both zero-curtain and cold season $CO_2$ emissions have increased, for one site of 0.17 and 0.36 gC m$^{-2}$ yr$^{-1}$ at Atqasuk, Alaska (Tao et al., 2021), and CLM5.0 simulations suggest that this is set to continue to 2100.





Future changes in shoulder season air temperature, soil moisture, and snow cover control both $CO_2$ and $CH_4$ due to moisture limitation, oxygen limitation and soil temperature. A major fraction of cold-season $CH_4$ emissions occur in the early winter shoulder season, particularly the zero-curtain (Zona et al., 2016) and interannual variability in both $CO_2$ and $CH_4$ depends largely on this period (Kittler et al., 2017). Soil moisture fluctuations (Figure 3) critically impact rates of microbial decomposition of organic matter via methanogenesis which favours wet, anoxic environments. $CH_4$ emissions are closely tied to soil moisture in the upper 30cm of soil and are closely correlated with soil moisture fluctuations during the soil freeze-in period (Sturtevant et al., 2011). In the early winter (Sep-Oct) the soil profile is saturated which drives anoxic conditions favoured by methanogens, which increases with depth (Arndt et al., 2020). Soil moisture increases in late September between RCP 4.5 and 8.5 (Figure 3) resulting in wetter soil, alongside increased zero-curtain period at depth (Figure 4) under RCP 8.5 present a risk of increased $CH_4$ emissions during this early winter period in the future at TVC. Such soil moisture increases, by slowing the soil freezing process have implications for the extension of elevated $CH_4$ emissions longer into the winter season (Sturtevant et al., 2012). Understanding the potential impacts of climate warming on both the early and late winter shoulder seasons is therefore key for assessing the risk of elevated winter carbon emissions from tundra soils. As well as persistent winter carbon emissions, 'pulses' of $CO_2$ and $CH_4$ have been observed during both the early and late winter shoulder seasons (Raz-Yaseef et al., 2017, Mastepanov et al., 2013) where trapped gasses are released as soils fluctuate between freeze and thaw. Soil emission pulses, which are not able to be simulated by CLM 5.0, are enhanced by an increase in rain-on-snow events. Such events accelerate soil warming in spring and soils are more susceptible to cracking and rapid gas release (Raz-Yaseef et al., 2017). A shift towards a rain dominated Arctic (Figure 2) may therefore increase the abundance of spring gas pulses from soils in both frequency and distribution, increasing cumulative carbon emissions annually. Further work is required to better understand the impact of these pulses on annual carbon budgets and their representation within LSMs.

**5 Conclusions**

Warming air and soil temperatures are facilitating higher rates of heterotrophic respiration in CLM5.0 simulations for TVC soils, resulting in an expected increase in winter carbon output in the future under RCP scenarios 4.5 and 8.5. Such winter emissions outpace those modelled by Natali et al. (2019) and therefore reinforce the vulnerability of Arctic soils and the carbon stored within to climate warming. This study has built on Callaghan et al. (2011) and quantified the effects of variability in snow timing and duration on future soil carbon emissions using CLM5.0.

Soil temperature simulations show a lengthening of the zero-curtain period in the future as well as deeper penetration of near-zero temperatures into the soil column which will mobilise legacy carbon in the future. A longer zero-curtain period, leading to a persistently unfrozen active layer of soil where higher rates of soil respiration persist for longer into the snow-cover period, poses a risk of significantly increased cumulative carbon output in the future from tundra soils.

Modelled winter carbon emissions demonstrate the importance of climate mitigation in preventing a significant increase in the Arctic winter carbon budget. Use of parameterisation for $K_{eff}$ from Sturm et al. (1997) reduces soil temperature biases



and presents significantly higher $CO_2$ production from 2016-2100 compared to default parameters from Jordan (1991). An improved representation of $K_{eff}$ exacerbates already increasing $CO_2$ emissions from Arctic tundra caused by a warming climate and the influence of the selected $K_{eff}$ parameterisation is shown to be as important, if not more, as the variability in future climate on simulated carbon emissions from Arctic tundra. Further work should aim to improve representations of snow thermal conductivity in LSMs while increasing the spatial coverage of future simulations to allow a more holistic

outlook for Arctic soil carbon emissions and better inform climate mitigation strategies and carbon budgets.

**Appendices**

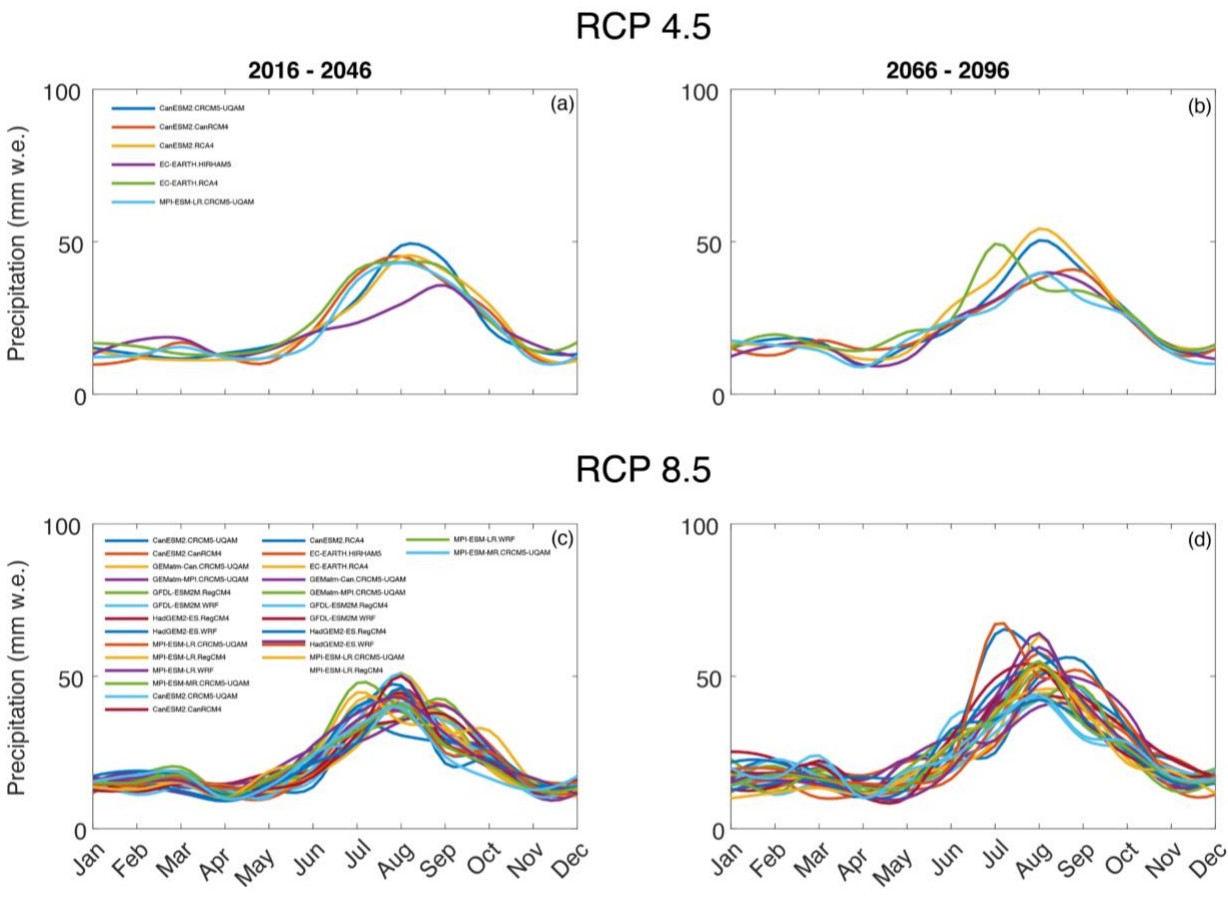

**APPENDIX A** – NA-CORDEX ensemble median monthly precipitation (Jan – Dec) split by individual ensemble member (RCP 4.5 n=6, RCP 8.5 n=27).

CLM5.0 simulations indicate minimal changes in late winter zero-curtain duration but a notable increase in early winter zero-curtain duration. The early winter has been identified as a key period in facilitating carbon emissions from Arctic tundra, particularly methane (Tao et al., 2021). Variation in future zero-curtain duration from August to December is presented below.



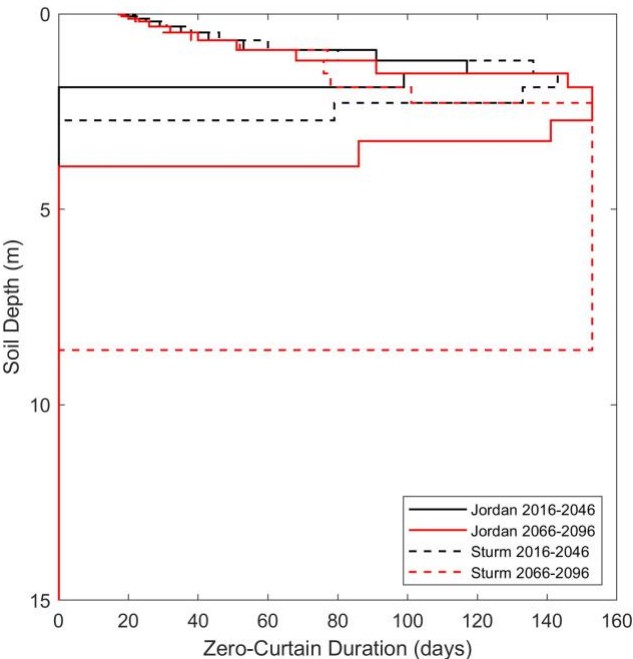

**APPENDIX B** – Aug-Dec zero-curtain duration with soil depth for CORDEX-Jordan (solid black and red lines) and CORDEX-Sturm
(dashed black and red lines) for 2016-2046 and 2066-2096.

To further constrain CLM5.0 parameter uncertainties values of $\Psi_{min}$ and Q10 were adjusted alongside two parameterisations
for $K_{eff}$ (Jordan and Sturm). A range of realistic $\Psi_{min}$ and Q10 values were chosen as per Dutch et al. (2023) alongside a range
of Q10ch4.

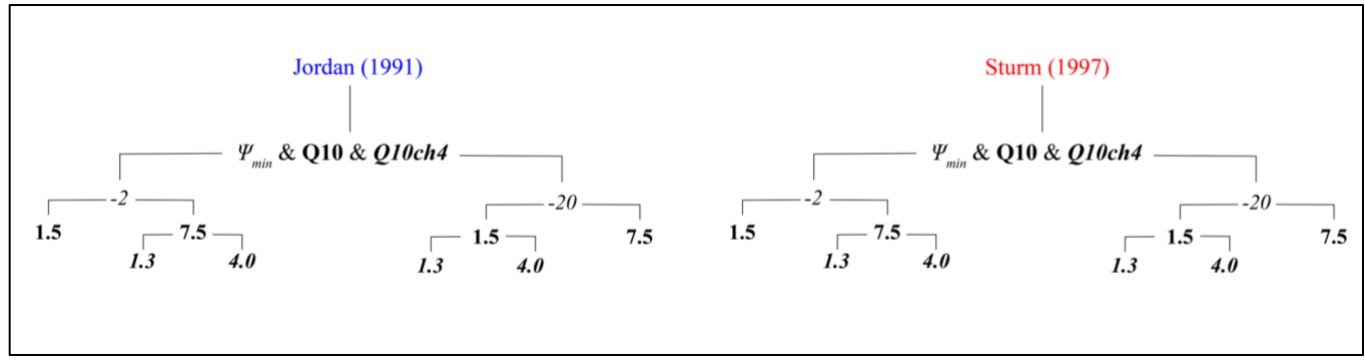

**APPENDIX C** – Schematic showing the range of Q10 (light blue) and $\Psi_{min}$ (green) values applied to CLM5.0 using CORDEX-Jordan (blue)
and CORDEX-Sturm (red) for the modelling experiment.

The full combination of applying changes in $K_{eff}$, $\Psi_{min}$, Q10 and Q10ch4 as well as the variability provided by the CORDEX
ensemble is presented in the main document. The below figure shows the upper and lower bounds of simulated $CO_2$ and $CH_4$
when parameter adjustments are applied, with Q10=7.5, $\Psi_{min}$ = -2 being the lower extreme and Q10=1.5, $\Psi_{min}$ = -20 being the
upper extreme.



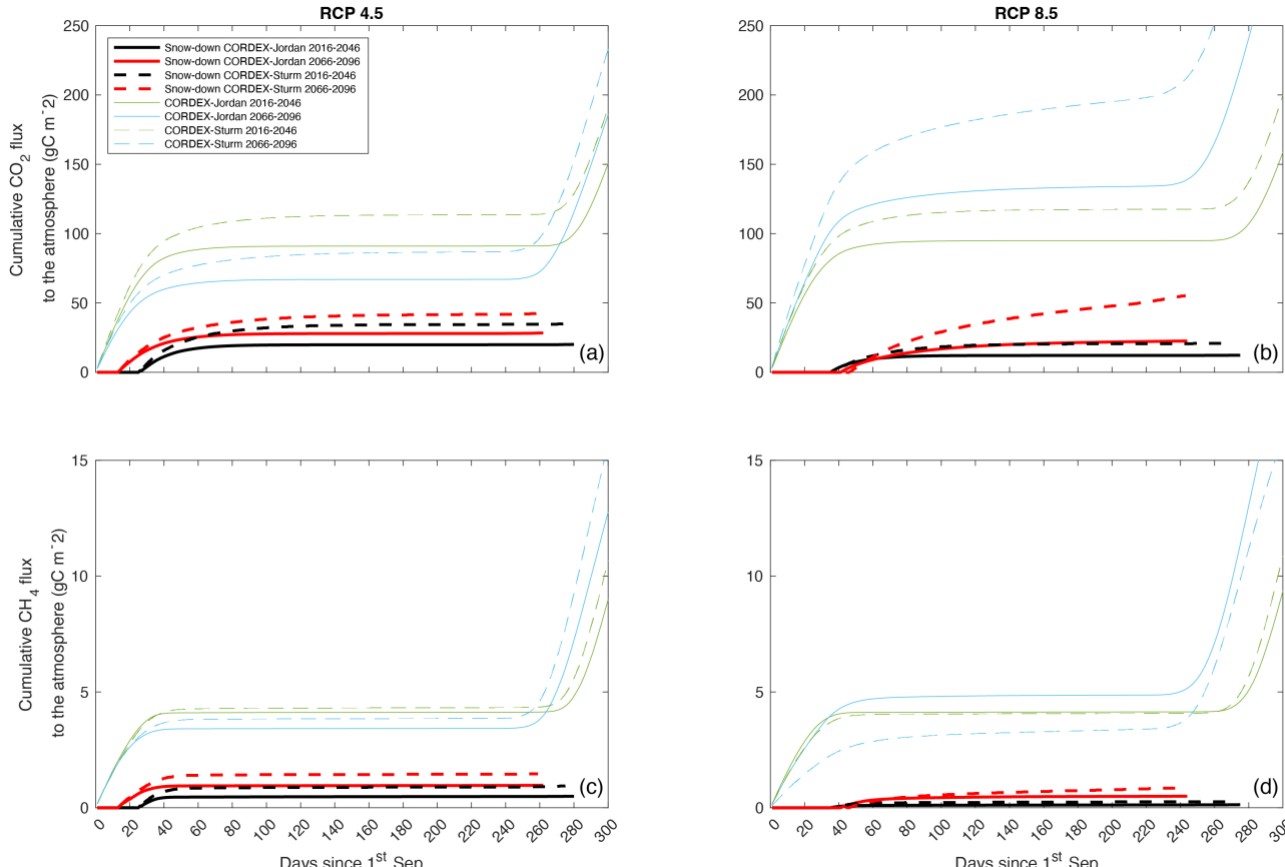

**APPENDIX D** – CLM5.0 simulated cumulative $CO_2$ and $CH_4$ flux to the atmosphere constrained by the snow-down period (red, black) and irrespective of snow (green, blue) over two 30-year time periods: 2016-2046 (black, green) and 2066-2096 (red, blue) under RCP 4.5 and RCP 8.5, for TVC using meteorological forcing from an ensemble of 33 RCM-GCM combinations (RC P 4.5 n=6, RCP 8.5 n=27). Solid lines show median values from CORDEX-Jordan ensemble and dashed lines show values for the CORDEX-Sturm ensemble. Variable plot line lengths are indicative of changes in snow-season length.







**APPENDIX E** – CLM5.0 simulated median soil respiration for the winter season, comparing the CORDEX-Jordan (blue) ensemble with CORDEX-Sturm (red) for two RCP scenarios (RCP 4.5 n=6, RCP 8.5 n=27). Shaded areas represent 75th and 25th percentiles and represent the CORDEX ensemble distribution. Each plot shows the extremes of Q10, $\Psi_{min}$ and Q10ch4 from the chosen parameter values seen in Appendix C, where the lower end $\Psi_{min}$ = -2, Q10 = 7.5, Q10ch4 = 4 supresses carbon output to and upper end $\Psi_{min}$ = -20, Q10 = 1.5, Q10ch4 = 1.3 which stimulates carbon output.

**Code and Data Availability**

Code and data to produce figures is available at: https://github.com/Jruthers/paper1/



## Author Contribution

Experimental Design, CORDEX Bias Correction, CLM5.0 Simulations, Analysis and Draft preparation; JR, NR, LW, AC. Supervision; NR, LW. All authors were involved in reviewing and editing prior to submission.

## Competing Interests

The authors declare no competing interests.

## Acknowledgements

JR was funded by a Research and Development Fund (RDF) Studentship from Northumbria University. The work was supported by a National Environmental Research Council (NERC) Seedcorn grant awarded to NR and LW; Carbon Emissions Under Arctic Snow (CEAS) project reference: NE/W003686/1. The authors thank R. Tutton, P. Marsh, R. Essery, R. Thorne, G. Hould Gosselin, B. Walker, O. Sonnentag, J. Griffith, and B. Darkin, for providing the field measurements required for CORDEX bias correction and thanks to Oliver Sonnentag and Bo Qu for advice on the approaches to process and bias correct CORDEX data. The authors thank C. Derksen, and G. Hould Gosselin for providing comments and edits to this manuscript. A raincloud plot function written by www.tomrmarshall.com was used to create the precipitation frequency plots shown in Figure 2.

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
