# Peer review of "Snow thermal conductivity controls future winter carbon emissions in shrubtundra"

_EGUsphere, 2024_

## Referee Comment (RC1)

**Summary:**

In their study, the authors evaluate the influence of a refined parameterization of snow effective thermal conductivity (Keff) on future wintertime $CO_2$ and $CH_4$ emissions from a shrub-tundra site in the Canadian Arctic. They show that using a Keff parameterization, that has in a previous study been found to better reproduce observed soil temperatures at the site, strongly enhances projections of wintertime $CO_2$ and $CH_4$ emissions for the second half of the 21st century under climate scenarios RCP 4.5 and RCP 8.5.

**Major comments:**

The study is highly relevant for future climate projections and its relevance is nicely stated in the manuscript: Snow cover plays an important role in regulating soil temperatures through thermal insulation. Changing snow properties such as snow density, wetness, and snow cover duration will affect the insulating function of snow which will influence soil temperatures and thus alter wintertime $CO_2$ and $CH_4$ production. It is therefore crucial to accurately represent the insulating property of snow in model simulations, that is to use a suitable parameterization of the snow effective thermal conductivity Keff. Further investigating the influence of snow on soil temperatures and greenhouse gas emissions might also help to explain higher observed than modelled wintertime carbon emissions.

Overall, the manuscript is written in a very concise and well-structured way that is easy to follow. Smooth transitions between the individual chapters further enhance the readability. The figures are well-designed and nicely support the findings described in the text.
One strength of the study its very clear focus – it investigates the effect of using a refined Keff parameterization on future wintertime greenhouse gas emissions. This focus is clearly stated in the title of the manuscript and I think that the manuscript could be further streamlined by strongly emphasizing this focus throughout the manuscript. This could be achieved by more strongly relating the findings on future developments (e.g. on the zero-curtain period) to the revised Keff parameterization. For this it would greatly help if the future development of Keff according to the Jordan and Sturm parameterizations could be shown explicitly – both their seasonal cycle (as shown for SWE, soil temperature, and soil moisture in Fig. 3) as well as their development over time (as shown for $CO_2$ and $CH_4$ emissions in Fig. 6). Simulated seasonal and interannual changes in Keff could then be discussed with respect to changes in the underlying snow properties which could potentially be related to the development of precipitation vs. temperature shown in Figure 2. More explicit discussion of future changes in Keff could help to relate previous findings, such as stated in ll. 288-299, to the findings obtained from this study.
More clearly focusing on the influence of Keff parameterization will further highlight the novelty of the study and make it stand out from the numerous studies that model future Arctic greenhouse gas emissions.

**Specific comments:**

Abstract:

l. 15: The transition from background information to the findings obtained in this study could be made clearer.

ll. 20-22: It would help if this statement could be elaborated a bit more in the main text or in the discussion section of the manuscript, if possible referring to one of the figures.

ll. 22-24: The changes in duration and penetration depth of the zero-curtain period could be more explicitly related to the revised parameterization of Keff.

ll. 24-26 The concluding statement should be more closely related to the role of snow and thereby to the title of the manuscript. As the current concluding remark is highly relevant as well, maybe a more specific statement could be added before.

Introduction:

The introduction could be streamlined and shortened by introducing future wintertime CO2 and CH4 emissions only once. Currently those are introduced in ll. 36-47 and in ll. 60-65.

ll. 67-74: This definition of "shoulder seasons" suggests a strong focus of on these transitional seasons throughout the manuscript. However, the reference to time periods around the snow cover period is not always clear in the results section. Maybe the season-specific role of snow could be clarified through showing and discussing the seasonal development of Keff and the underlying snow properties.

ll. 75-91: Here you could highlight even more the novel contribution of your study – it builds on the findings by Dutch et al. (2023) who showed that the Sturm Keff parameterization better reproduces observed soil temperatures at TVC. In your study, you now investigate the influence of this revised Keff parameterization of future projections of greenhouse gas emissions from TVC.

ll. 90-91: The findings on the sensitivity of CO2 and CH4 fluxes to $\Psi_{min}$, Q10, and Q10ch4 are not mentioned in the main text although the parameters are mentioned both here in the study aim as well as explained in the methods section. Would it be possible to state a key finding related to these parameters in the main text and refer to Appendix E?

Methodology:

Section 2.3: Definitions of $h_{sl}$ and $r_W$ are missing. Also, abbreviations $\rho$, $K_{air}$, and $K_{ice}$ could be introduced explicitly.

Results:

ll. 192-193: Figure 2 shows shoulder season conditions only. Did you produce these graphs also for other months and could they potentially be included as an appendix figure to prove your statement that the shift in precipitation from snow to rain is most pronounced in the shoulder seasons?

l. 195: Figure 2 is very well-designed and informative. As it contains a lot of information a very clear description of what exactly is shown in the figure would be helpful. To me the variable of "precipitation frequency" was not intuitive at first. Maybe you could replace the term "median monthly precipitation" in the figure caption and explain instead that the graphs represent a frequency distribution of precipitation events related to the air temperatures.

Section 3.2: This section could be split into two subsections (3.2.1 and 3.2.2): One on the simulations of environmental conditions (SWE, soil temperature, soil moisture; ll. 210-243) and one on the simulated CO2 and CH4 emissions (ll. 245 - 279).

l. 216: Either "where" or "because" needs to be removed.

l. 222: Reference error.

l. 240: While one could argue that they would fit better into the discussion section of the manuscript, I very much like the smooth transitions between the individual chapters such as in ll. 206-208, 241-244. However, some additional interpretations that do not directly serve as transitional sentences could be moved from the results to the discussion section. Those include ll. 240-241, 253-255, and 261-262.

ll. 249-259 & Fig. 5: The results shown here are highly interesting and relevant. However, I find their description in the text a little confusing.
In l. 250, do you mean an "earlier onset of snow"? If I understand correctly, looking at Figure 3, the onset of snow is earlier in both RCP 4.5 than in RCP 8.5 and earlier in 2066-2096 than 2016-2046 for RCP 4.5.
In Figure 5, are CO2 and CH4 emissions zero before the onset of snow or were emissions not simulated for the snow-free period? In the latter case, I would prefer if the graphs were to start only with the onset of snow.
As you describe in ll. 251-252, cumulative carbon emissions during the snow-cover period are reduced under RCP 8.5 compared to RCP 4.5 due to a delayed snow onset. At first, this seems contradictory to the higher simulated future total carbon emissions (Figure 6). In ll. 262-265 you refer to Appendix D to clarify the matter. However, this explanation is not entirely clear to me. Maybe it would help to more explicitly describe the relation between snow-cover season and annual carbon emissions and their future developments. This would also further highlight the importance of early winter emissions.
It should furthermore be clarified what "unconstrained by snow cover" and "irrespective of snow" means in both l. 263 and in the figure caption of Appendix D. Does it mean that the snow cover is still considered in the simulation (as indicated by the effect of the Keff parameterization) and that carbon emissions are simulated for the entire 300 days following September 1$^{st}$ instead of for the snow-covered period only?
The findings from Figure 5 should be discussed in the discussion section, referring to the seasonal change in Keff and soil temperatures under the different scenarios and parameterizations.

Discussion

I think that the entire discussion section could benefit from a stronger focus on your specific findings and on the effect of snow properties as opposed to the more general speculations on future changes in snow cover and carbon emissions.
For example, do your model simulations support the previous finding of increasing Keff in the future (l. 295)?
Similarly, changes in the zero-curtain period should be discussed with a more explicit relation to the parameterization of snow properties (ll. 320-332).

Conclusions

l. 357: You could consider referring to Callaghan et al. (2011) already in the introduction to stress its high relevance for your study.

l. 357-358: The overall study aim here sounds different to me than the one stated in ll. 90 – 91. I would suggest keeping the main focus on the snow thermal conductivity.

---

## Author Response (AR1)

**Initial Author Response for "Snow thermal conductivity controls future winter carbon emissions in shrub-tundra", Rutherford *et al*.**

The authors would like to thank the editor and reviewers for taking the time to read and review the manuscript and for the suggested improvements. The reviewer comments are included here in **black** and our initial responses are in green. Where we refer to line numbers in our response, this relates to the original submitted document.

**Reply to open discussion comment 1:**

This is a very important paper and set of discoveries.

However, it is missing recognition of the massive set of studies that discovered and addressed these issues going back to the original US ITEX program at Toolik AK and the NSF ATLAS (Arctic Transition in the Land Atmosphere System) that should be included.

Those being:

Jones, M. H., Fahnestock, J. T., Walker, D. A., Walker, M. D., and Welker, J. M. (1998) Carbon dioxide fluxes in moist and dry arctic tundra during the snow-free season: responses to increases in summer temperature and winter snow accumulation. Arctic and Alpine Research 30: 373-380.

Fahnestock, J. T., Jones, M. H., Brooks, P. D., Walker, D. A., and Welker, J. M. (1998) Winter and early spring CO2 flux from tundra communities of northern Alaska. Journal of Geophysical Research 102 (D22): 29925-29931.

Fahnestock, J. T., Jones, M. H., Brooks, P. D., and Welker, J. M. (1999) Significant CO2 emissions from tundra soils during winter: Implications for annual carbon budgets of arctic communities. Global Biogeochemical Cycles 13: 775-779.

Welker, J. M., Fahnestock, J. T., and Jones, M. H. (2000) Annual CO2 flux from dry and moist arctic tundra: Field responses to increases in summer temperature and winter snow depth. Climatic Change 44: 139-150.

Schimel, J. S., Bilbrough, C. B., and Welker, J. M. (2004) Increased snow depth affects microbial activity and nitrogen mineralization in two Arctic tundra communities. Soil Biology and Biochemistry 36: 217-227.

Schimel, J., Fahnestock, J., Michaelson, G., Milkan, C., Ping, C, Romanovsky, V., and Welker, J. M. (2006) Cold-season production of CO2 in Arctic soils: Can laboratory and field estimates be reconciled through a simple modeling approach? Arctic, Antarctic and Alpine Research 38(2): 249-255

Sullivan, P. F., Arens, S., Sveinbjörnsson, B., and Welker, J. M. (2010) Modeling the seasonality of belowground respiration along an elevation gradient in the western Chugach Mountains, Alaska. Biogeochemistry 101(1-3): 61-75.

Lupascu, M., Czimczik, C. I., Welker, M., Cooper, L., and Welker, J. M. (2018) Winter ecosystem respiration and sources of CO2 from the High Arctic tundra of Svalbard: Response to a deeper snow experiment. JGR Biogeosciences DOI.org/10.1029/ 2018JG004396.

Pedron, S., Xu, X., Walker, J., Welker, J. M., Klein, E. and Czimczik, C. (2021) Time-integrated Collection of CO2 for 14C Analysis from Soils. Radiocarbon DOI: 10.101/RDC.2021.42.

Pedron, S. A., Welker, J. M., Euskirchen, E., Klein, E. S., Walker, J. C., Xu, X., and Czimczik, C. I. (2022) Closing the winter gap-Year-round measurements of soil CO2 emission sources in Arctic Tundra. Geophysical Research Letters doi.org/10.1029/2021GL097347.

The authors would like to thank the reviewer for their interest and engagement in our paper. As a result of the comments received, we include the following additions:

In the Introduction:
-       Fahnestock, J. T., Jones, M. H. & Welker, J. M. 1999. Wintertime CO2 efflux from arctic soils: implications for annual carbon budgets. Global Biogeochemical Cycles, 13, 775-779.
-       Welker, J. M., Fahnestock, J. T., and Jones, M. H. (2000) Annual CO2 flux from dry and moist arctic tundra: Field responses to increases in summer temperature and winter snow depth. Climatic Change 44: 139-150.
-       Schimel, J., Fahnestock, J., Michaelson, G., Milkan, C., Ping, C, Romanovsky, V., and Welker, J. M. (2006) Cold-season production of CO2 in Arctic soils: Can laboratory and field estimates be reconciled through a simple modeling approach? Arctic, Antarctic and Alpine Research 38(2): 249-255

We add the following text:
"Arctic tundra ecosystems, once considered to be carbon sinks, are increasingly acting as carbon sources due to elevated winter carbon emissions driven by rising temperatures and deeper snow cover (Pongracz et al., 2021, Fahnestock et al., 1999, Welker et al., 2000, Schimel et al., 2004)".

In the Discussion:
-       Pedron, S., Jespersen, R., Xu, X., Khazindar, Y., Welker, J. & Czimczik, C. 2023. More snow accelerates legacy carbon emissions from arctic permafrost. AGU Advances, 4, e2023AV000942.

Pedron et al. (2023) already forms part of the discussion regarding the mobilization of legacy carbon within Arctic permafrost.

-       Schimel, J., Fahnestock, J., Michaelson, G., Milkan, C., Ping, C, Romanovsky, V., and Welker, J. M. (2006) Cold-season production of CO2 in Arctic soils: Can laboratory and field estimates be reconciled through a simple modeling approach? Arctic, Antarctic and Alpine Research 38(2): 249-255

We add the following text:
"The seasonality of soil temperature is critical in controlling winter $CO_2$ emissions from soils, particularly the length of time at which soil remains at or near 0°C at the beginning of winter (Schimel et al., 2006)."

**Reply to comments from Reviewer 1:**

**Major comments:**

The study is highly relevant for future climate projections and its relevance is nicely stated in the manuscript: Snow cover plays an important role in regulating soil temperatures through thermal insulation. Changing snow properties such as snow density, wetness, and snow cover duration will affect the insulating function of snow which will influence soil temperatures and thus alter wintertime CO2 and CH4 production. It is therefore crucial to accurately represent the insulating property of snow in model simulations, that is to use a suitable parameterization of the snow effective thermal conductivity Keff. Further investigating the influence of snow on soil temperatures and greenhouse gas emissions might also help to explain higher observed than modelled wintertime carbon emissions.

Overall, the manuscript is written in a very concise and well-structured way that is easy to follow. Smooth transitions between the individual chapters further enhance the readability. The figures are well-designed and nicely support the findings described in the text.

Thank you for your positive feedback on the manuscript's relevance and structure, we are pleased that the importance of accurately representing snow thermal conductivity and its impacts are clearly conveyed.

One strength of the study its very clear focus – it investigates the effect of using a refined Keff parameterization on future wintertime greenhouse gas emissions. This focus is clearly stated in the title of the manuscript and I think that the manuscript could be further streamlined by strongly emphasizing this focus throughout the manuscript.

This could be achieved by more strongly relating the findings on future developments (e.g. on the zero-curtain period) to the revised Keff parameterization. For this it would greatly help if the future development of Keff according to the Jordan and Sturm parameterizations could be shown explicitly – both their seasonal cycle (as shown for SWE, soil temperature, and soil moisture in Fig. 3) as well as their development over time (as shown for CO2 and CH4 emissions in Fig. 6).

Simulated seasonal and interannual changes in Keff could then be discussed with respect to changes in the underlying snow properties which could potentially be related to the development of precipitation vs. temperature shown in Figure 2. More explicit discussion of future changes in Keff could help to relate previous findings, such as stated in ll. 288-299, to the findings obtained from this study.

More clearly focusing on the influence of Keff parameterization will further highlight the novelty of the study and make it stand out from the numerous studies that model future Arctic greenhouse gas emissions.

**We will follow the suggestion of the reviewer and visualize K$_{eff}$ as modelled by CLM5.0 under Jordan (1991) and Sturm et al. (1997) parameterisations to 2100. We will show K$_{eff}$ seasonally across two time frames: 2016-2046 and 2066-2096 which presents K$_{eff}$ over the seasonal cycle as well as demonstrating its development over time. This visualization will form part of the refined Figure 3:**

[Figure]

**Figure 3** – CLM5.0 simulated median daily snow water equivalent (SWE; a,b), soil liquid water (12cm) content (c,d) snow thermal conductivity (e,f) and 10cm soil temperature (g,h) over two 30-year time periods: 2016-2046 (black) and 2066-2096 (red) under RCP 4.5 and RCP 8.5, for TVC using input meteorological data from an ensemble of 33 RCM-GCM combinations (RCP 4.5 n=6, RCP 8.5 n=27). Solid and dashed lines show ensemble median values for CORDEX-Jordan and CORDEX-Sturm experiments respectively.

**We add the following text to the description of Figure 3 results:**

**"Compared to CORDEX-Jordan, Snow thermal conductivity using CORDEX-Sturm is lower on average by 0.07 and 0.14 W m-1 K-1 under RCP 4.5 and RCP 8.5 respectively."**

**Specific comments:**

Abstract:
l. 15: The transition from background information to the findings obtained in this study could be made clearer.

**Added text: "To address this, we investigated the impacts of implementing a $K_{eff}$ parameterisation more suitable to Arctic snowpacks into the Community Land Model (CLM5.0)."**

ll. 20-22: It would help if this statement could be elaborated a bit more in the main text or in the discussion section of the manuscript, if possible referring to one of the figures.

**We will emphasise the impacts of the refined $K_{eff}$ throughout the manuscript as per other comments below.**

ll. 22-24: The changes in duration and penetration depth of the zero-curtain period could be more explicitly related to the revised parameterization of Keff.

**This section of the abstract is refined as follows:**

**"Furthermore, CLM5.0 simulations using the refined $K_{eff}$ show an extension of the early winter (Sept-Oct) zero-curtain, by nearly a month under RCP 8.5. Consequently, recent increases in both zero-curtain duration and winter $CO_2$ emissions appear set to continue to 2100."**

ll. 24-26 The concluding statement should be more closely related to the role of snow and thereby to the title of the manuscript. As the current concluding remark is highly relevant as well, maybe a more specific statement could be added before.

**Added text: "The average difference in refined $K_{eff}$ compared with the default $K_{eff}$ raises minimum winter soil temperatures by 4-7 °C by the end of the century under RCP 4.5 and 8.5."**

Introduction:

The introduction could be streamlined and shortened by introducing future wintertime CO2 and CH4 emissions only once. Currently those are introduced in ll. 36-47 and in ll. 60-65.

**Lines 61-65 are removed to be more concise.**

ll. 67-74: This definition of "shoulder seasons" suggests a strong focus of on these transitional seasons throughout the manuscript. However, the reference to time periods around the snow cover period is not always clear in the results section. Maybe the season-specific role of snow could be clarified through showing and discussing the seasonal development of Keff and the underlying snow properties.

**Our current focus on the shoulder seasons in the manuscript are as follows:**

1. **The examination of shoulder season months (April, May, September, October) in Figure 2.**

2. **Visualisation and discussion of shoulder season soil temperature, the zero-curtain period (Figure 3&4).**

3. **The impact of shoulder season emissions on the overall winter budget (Figure 5).**

**To build on this we will visualize the seasonal development of $K_{eff}$ to reinforce the impact of snow on the seasonal cycle of soil temperature and carbon fluxes, and add the following text to demonstrate**

**emphasis on shoulder season effects:**

**"This study addresses this knowledge gap by examining how projected changes in shoulder season air temperature, precipitation, snow thermal conductivity and soil temperature influence cold season carbon dynamics (Figures 2-5)."**

**"In shoulder season months (April, May, September and October), projected air temperatures increase on average by 1.6°C under RCP 4.5 and 3.7°C under RCP 8.5."**

**"In the future under RCP 8.5, autumn shoulder season (Sep – Oct) near-zero temperatures penetrate up to 6m deeper into the soil column compared with the present day in the CORDEX-Sturm simulations (Figure 4h)."**

**"By the start of November, cumulative $CO_2$ emissions have reached 50-90% of their winter totals, highlighting the importance of autumn shoulder season emissions to the winter $CO_2$ budget. Under RCP 4.5 there is minimal increase in cumulative $CO_2$ after mid-late November (after day 80) with increases of 0.02 g$CO_2$ m$^{-2}$ day$^{-1}$ and 0.00015 g$CH_4$ m$^{-2}$ day$^{-1}$. Comparatively, under RCP 8.5, higher levels of emissions continue deeper into the winter with average increases of 0.07 g$CO_2$ m-2 day-1 and 0.003 g$CH_4$ m$^{-2}$ day$^{-1}$ (day 80 to day 240) (Figure 5)."**

ll. 75-91: Here you could highlight even more the novel contribution of your study – it builds on the findings by Dutch et al. (2023) who showed that the Sturm Keff parameterization better reproduces observed soil temperatures at TVC. In your study, you now investigate the influence of this revised Keff parameterization of future projections of greenhouse gas emissions from TVC.

**We now include the following:**

**"Our study applies Q10 and $\Psi_{min}$ values suitable for tundra soil as defined by Dutch et al. (2023) to investigate the influence of different $K_{eff}$ parameterisations on future projections of soil temperature and carbon emissions."**

ll. 90-91: The findings on the sensitivity of CO2 and CH4 fluxes to Ψmin, Q10, and Q10ch4 are not mentioned in the main text although the parameters are mentioned both here in the study aim as well as explained in the methods section. Would it be possible to state a key finding related to these parameters in the main text and refer to Appendix E?

**We now display the mean difference between carbon fluxes under different values $\Psi_{min}$, Q10, and Q10ch4 from 2016-2100 in Appendix E1:**

[Figure]

**APPENDIX E1 –** CLM5.0 simulated median soil respiration for the winter season, comparing the CORDEX-Jordan (blue) ensemble with CORDEX-Sturm (red) for two RCP scenarios (RCP 4.5 n=6, RCP 8.5 n=27). Shaded areas represent 75th and 25th percentiles and represent the CORDEX ensemble distribution. Each plot shows the extremes of Q10, $\Psi_{min}$ and Q10ch4 from the chosen parameter values seen in Appendix C1, where the lower end $\Psi_{min}$ = -2, Q10 = 7.5, Q10ch4 = 4 supresses carbon output to and upper end $\Psi_{min}$ = -20, Q10 = 1.5, Q10ch4 = 1.3 which stimulates carbon output. The average difference between the displayed parameters (solid versus dashed lines, $\Delta$) is included on each subplot.

**We also add the following text to the results:**

**"Simulations with reduced $\Psi_{min}$ and (-20) consistently stimulate higher carbon fluxes under RCP 4.5 and 8.5 compared to simulations using the default $\Psi_{min}$ value (-2) (Appendix E1). This is particularly evident under CORDEX-Sturm which shows average increases of 0.13-0.16 gCO$_2$ m$^{-2}$ day$^{-1}$ and 0.0014-0.0018 gCH$_4$ m$^{-2}$ day$^{-1}$ from 2016-2100."**

**We also add the following text to the discussion:**

**"Soil moisture and temperature are critical controls of soil carbon emissions and adjustments to $\Psi_{min}$, Q10 and Q10ch4 bring CLM5.0 simulations into closer alignment with field measurements (Dutch et al., 2023). Future CO$_2$ and CH$_4$ emissions show greater seasonal variability under CORDEX-Sturm compared with CORDEX-Jordan, particularly under RCP 8.5, which suggests that soil moisture and thermal dynamics are more sensitive to snow cover in the CORDEX-Sturm configuration (Appendix E). "**

Methodology:

Section 2.3: Definitions of $h_{sl}$ and $r_W$ are missing. Also, abbreviations $\rho$, $K_{air}$, and $K_{ice}$ could be introduced explicitly.

**We now include explicit definitions of these variables in section 2.3.**

Results:

ll. 192-193: Figure 2 shows shoulder season conditions only. Did you produce these graphs also for

other months and could they potentially be included as an appendix figure to prove your statement that the shift in precipitation from snow to rain is most pronounced in the shoulder seasons?

**We now include a visualisation of March, June, August and November as 'Appendix F' which shows all precipitation events falling either as snow or rain. This visualization supports our choice of visualizing April, May, September, October in the main body of the manuscript, which show shifts in precipitation phase from present to future.**

l. 195: Figure 2 is very well-designed and informative. As it contains a lot of information a very clear description of what exactly is shown in the figure would be helpful. To me the variable of "precipitation frequency" was not intuitive at first. Maybe you could replace the term "median monthly precipitation" in the figure caption and explain instead that the graphs represent a frequency distribution of precipitation events related to the air temperatures.

**We have revised the caption of Figure 2 as follows:**

**"Half-violin plots show the frequency distribution of precipitation events as a function of air temperature for TVC in April, May, September and October under RCP 4.5 (left) and RCP 8.5 (right). from an ensemble of 33 NA-CORDEX GCM-GCM combinations (RCP 4.5 n=6, RCP 8.5 n=27). White violins with black outlines represent 2016-2046 and red violins 2066-2096. The height of each violin represents the frequency of precipitation events at a given temperature. The solid and dashed black lines at 0 and 2 °C show transitional temperatures between snow and rain where CLM5.0 treats the transition in precipitation phase as a linear ramp. Inset boxplots show monthly total precipitation for the two 30-year periods."**

Section 3.2: This section could be split into two subsections (3.2.1 and 3.2.2): One on the simulations of environmental conditions (SWE, soil temperature, soil moisture; ll. 210-243) and one on the simulated $CO_2$ and $CH_4$ emissions (ll. 245 - 279).

**These new subsections are now included: 3.2.1 Environmental conditions, 3.2.2 $CO_2$ and $CH_4$ emissions.**

l. 216: Either "where" or "because" needs to be removed.

**"Because" is removed.**

l. 222: Reference error.

**This cross-reference error is resolved.**

l. 240: While one could argue that they would fit better into the discussion section of the manuscript, I very much like the smooth transitions between the individual chapters such as in ll. 206-208, 241-244. However, some additional interpretations that do not directly serve as transitional sentences could be moved from the results to the discussion section. Those include ll. 240-241, 253-255, and 261-262.

**Line 240-241 has been moved to the discussion and modified as follows:**

**"Such increases in soil temperature and zero-curtain duration demonstrates both the influence of snow on soil temperatures at depth and the risks of climate warming on permafrost degradation and possible mobilisation of legacy carbon from Arctic soils."**

**Line 253-255 and 261-262 are removed as these points repeat text elsewhere in the discussion.**

ll. 249-259 & Fig. 5: The results shown here are highly interesting and relevant. However, I find their description in the text a little confusing.

**We have read through lines 249-259 and anticipate that the sections the reviewer is referring to are these:**

**[1] – "which suggests that the contribution of early winter emissions to the overall accumulation of winter emissions is key, in the period where the snowpack is accumulating and soil temperatures are cooling."**

**And**

**[2] – "The importance of the early winter autumn shoulder season $CO_2$ release to the overall carbon budget under RCP 4.5 is clear as there is minimal increase in cumulative carbon after mid-late November with increases of 0.02 $gCO_2$ $m^{-2}$ $day^{-1}$ and 0.00015 $gCH_4$ $m^{-2}$ $day^{-1}$ (day 80 to day 240) averaged across present/future and CORDEX-Jordan/Sturm. By contrast, under RCP 8.5, $CO_2$ emissions are prolonged into the winter due to warmer soil temperatures, with an average increase of 0.07 $gCO_2$ $m^{-2}$ $day^{-1}$ and 0.003 $gCH_4$ $m^{-2}$ $day^{-1}$ (day 80 to day 240) (Figure 5)."**

**We combine and amend these sections as follows:**

**[1] - This sentence is removed.**

**We then add the following text:**

**"By the start of November, cumulative $CO_2$ emissions have reached 50-90% of their winter totals, highlighting the importance of autumn shoulder season emissions to the winter $CO_2$ budget. Under RCP 4.5 there is minimal increase in cumulative $CO_2$ after mid-late November (after day 80) with increases of 0.02 $gCO_2$ $m^{-2}$ $day^{-1}$ and 0.00015 $gCH_4$ $m^{-2}$ $day^{-1}$. Comparatively, under RCP 8.5, higher levels of emissions continue deeper into the winter with average increases of 0.07 $gCO_2$ $m^{-2}$ $day^{-1}$ and 0.003 $gCH_4$ $m^{-2}$ $day^{-1}$ (day 80 to day 240) (Figure 5)."**

In l. 250, do you mean an "earlier onset of snow"? If I understand correctly, looking at Figure 3, the onset of snow is earlier in both RCP 4.5 than in RCP 8.5 and earlier in 2066-2096 than 2016-2046 for RCP 4.5.

**This is now corrected to "earlier".**

In Figure 5, are CO2 and CH4 emissions zero before the onset of snow or were emissions not simulated for the snow-free period? In the latter case, I would prefer if the graphs were to start only with the onset of snow.
**Cumulative carbon fluxes are plotted from 1$^{st}$ September to allow intercomparison across time periods in which snow season length may vary (i.e. the start date of snow accumulation >5mm**

**SWE).**

**We specify the constraints of Figure 5 in the Methodology:**
**"A major focus of this study is the winter season when snow is on the ground (i.e. the snow-covered non-growing season). We define this period as the time when all model ensemble members (RCP 4.5 n=6, RCP 8.5 n=27) agree that SWE is >5mm. Simulations of SR and FCH4 are filtered by these constraints so the analysis in Figures 5 and 6 is focused only on carbon fluxes across a common snow-covered, non-growing season in all scenarios and forcing datasets."**

As you describe in ll. 251-252, cumulative carbon emissions during the snow-cover period are reduced under RCP 8.5 compared to RCP 4.5 due to a delayed snow onset. At first, this seems contradictory to the higher simulated future total carbon emissions (Figure 6). In ll. 262-265 you refer to Appendix D to clarify the matter. However, this explanation is not entirely clear to me. Maybe it would help to more explicitly describe the relation between snow-cover season and annual carbon emissions and their future developments. This would also further highlight the importance of early winter emissions.

It should furthermore be clarified what "unconstrained by snow cover" and "irrespective of snow" means in both l. 263 and in the figure caption of Appendix D. Does it mean that the snow cover is still considered in the simulation (as indicated by the effect of the Keff parameterization) and that carbon emissions are simulated for the entire 300 days following September $1^{st}$ instead of for the snow-covered period only?

**Our CLM5.0 simulations are year-round, however the visualised accumulation of emissions in Figure 5 are constrained by snow-down (where SWE >5mm). Hence, in the future under RCP 8.5 due to warming causing a shorter snow-covered season, the overall accumulated emissions are lower compared to the longer snow season in RCP 4.5. Appendix D1 aims to reinforce the importance of the early snow-season in the total accumulation of winter emissions. Blue and green lines in Appendix D1 show $CO_2$ and $CH_4$ cumulative emissions irrespective of whether snow is on the ground therefore containing periods of snow-free emissions as well as snow-covered emissions. This contrasts with Figure 5 which only shows snow-covered emissions.**

**To help clarify these visualizations we make the following amendments to the text:**

**"Appendix D1 presents a comparison between cumulative emissions limited to when snow is on the ground, as per Figure 5, and those unconstrained by snow cover."**

The findings from Figure 5 should be discussed in the discussion section, referring to the seasonal change in Keff and soil temperatures under the different scenarios and parameterizations.

**We add the following comments to the discussion to focus on the influence of $K_{eff}$ on accumulated emissions:**

**"Under RCP 8.5 cumulative winter carbon emissions from soils at TVC are projected to increase in the future despite a reduction in snow-cover duration. A reduced $K_{eff}$, as introduced by CORDEX-Sturm (Figure 3), increases cumulative winter $CO_2$ and $CH_4$ emissions by 50-150% compared with CORDEX-Jordan under both RCP 4.5 and 8.5".**

Discussion

I think that the entire discussion section could benefit from a stronger focus on your specific findings and on the effect of snow properties as opposed to the more general speculations on future changes in snow cover and carbon emissions.

For example, do your model simulations support the previous finding of increasing Keff in the future (l. 295)?

**Lines 294-297 are removed as they are speculative and don't align with our visualization of $K_{eff}$.**

**We add emphasis to the refinement of $K_{eff}$ in the discussion through addressing previous comments by:**

- **Discussing sensitivity of parameters to $\Psi_{min}$, Q10 and Q10ch4 under CORDEX-Jordan and CORDEX-Sturm.**
  - **"Soil moisture and temperature are critical controls of soil carbon emissions and adjustments to $\Psi_{min}$, Q10 and Q10ch4 bring CLM5.0 simulations into closer alignment with field measurements (Dutch et al., 2023). Future $CO_2$ and $CH_4$ emissions show greater seasonal variability under CORDEX-Sturm compared with CORDEX-Jordan, particularly under RCP 8.5, which suggests that soil moisture and thermal dynamics are more sensitive to snow cover in the CORDEX-Sturm configuration (Appendix E1). "**

- **Including explicit impacts of snow representation on zero-curtain duration.**
  - **"Simulations performed in this study indicate an increase in both the duration and depth of the early winter zero curtain under future climate conditions (Figure 4). Under CORDEX-Sturm, the projected early winter zero curtain extends up to 26 days longer and reaches depths up to 6m deeper than under CORDEX-Jordan further highlighting the impact of snow representation on simulated soil temperatures."**

- **Reference to the new visualisation of $K_{eff}$ when discussing future cumulative carbon flux simulations.**
  - **"A reduced $K_{eff}$ (0.07-0.14 W m$^{-1}$ K$^{-1}$), as introduced by CORDEX-Sturm (Figure 3), increases cumulative winter $CO_2$ and $CH_4$ emissions from 2016-2100 by 50-150% compared with CORDEX-Jordan under both RCP 4.5 and 8.5."**

Similarly, changes in the zero-curtain period should be discussed with a more explicit relation to the parameterization of snow properties (ll. 320-332).

**We amend this discussion point and better clarify the impact of $K_{eff}$ refinement and future climate on projected soil temperatures.**

**"Simulations performed in this study indicate an increase in both the duration and depth of the early winter zero curtain under future climate conditions (Figure 4). Under CORDEX-Sturm, the projected zero-curtain duration extends up to 26 days longer than under CORDEX-Jordan further highlighting the impact of snow representation on simulated soil temperatures."**

Conclusions

l. 357: You could consider referring to Callaghan et al. (2011) already in the introduction to stress its high relevance for your study.

**To further refer to this paper, and to reinforce the role of Arctic snow we add the following to the introduction:**

**"Arctic snow is a key determinant of ground temperature and plays a major role in the wider hydrological and ecological Arctic system (Callaghan et al., 2011)."**

l. 357-358: The overall study aim here sounds different to me than the one stated in ll. 90 – 91. I would suggest keeping the main focus on the snow thermal conductivity.

**Lines 357-398 are removed and we add emphasis on $K_{eff}$ with the following text:**

**"Projected $CO_2$ and $CH_4$ emissions are highly sensitive to parameters $K_{eff}$, $\Psi_{min}$, Q10 and Q10ch4 which govern soil respiration. We find that lower $\Psi_{min}$ consistently increases cold season carbon**

fluxes and higher Q10 suppressed them, which aligns with the findings of Dutch et al. (2023). Implementing the Sturm et al. (1997) $K_{eff}$ parameterisation increased the sensitivity of modelled carbon emissions to $\Psi_{min}$ and Q10 compared with the default $K_{eff}$ parameterisation.

**Reply to comments from Reviewer 2:**

Rutherford and others study the consequences of using more realistic snow thermal parameters and biogeochemical temperature sensitivities for future carbon dioxide and methane release in a tundra ecosystem. The results make some interesting points but I was unsure why the particular site was chosen if no data are being compared against model results, especially as the earlier simulation period overlaps with the present day. As such it is unclear if the base model is realistic in the first place, which is critical for defensible future scenarios. I recommend trying to use existing observations, especially for things like snow duration and soil temperature that are measurable, for ensuring that model results are realistic before moving on to the important topic of making the model more realistic.

The authors would like to thank the reviewer for their interest in our manuscript and for highlighting the importance of model validation. Our choice of Trail Valley Creek (TVC) is grounded in its extensive observational dataset beginning in the early 1990s, which has made it the focus of many measurement and modelling studies. Notably, Dutch et al. (2023), focusses on CLM5.0 simulations at TVC and show that implementing the Sturm et al. (1997) $K_{eff}$ parameterisation alleviates cold soil biases in CLM5.0 simulations when compared with field measurements. Further, use of the Sturm et al. (1997) parameterisation brings NEE simulations more closely in line with field measurements. In our manuscript we use the work by Dutch et al. (2023) to provide the essential scientific foundations the reviewer requests, which shows CLM5.0 configurations to be realistic. The process of showing the model is realistic is a paper in itself, i.e. Dutch et al. (2023). Here we leverage these realistic parameterisations to provide more robust projections of soil and carbon.

To address this, we have added the following text to the introduction to reinforce the performance of the model against field observations of soil temperature and net ecosystem exchange (NEE) as outlined in Dutch et al. (2023):

We add the following text to the introduction:
"Implementing the Sturm et al. (1997) parameterisation reduces the cold bias in simulated soil temperatures by two-thirds. Further, while default CLM5.0 produces negligible winter NEE, combining the Sturm et al. (1997) parameterisation with a mid-range value for $\Psi_{min}$ (-20) produces winter NEE values consistent with field observations (Dutch et al. 2023, Figure 5)"

We add the following text to the discussion:
"The CLM5.0 parameterisations for Keff, Q10 and $\Psi$min explored by Dutch et al. (2023) were found to be highly suitable for representing winter soil temperatures and carbon fluxes under present day conditions. This alignment between observations and simulations provides confidence in the model's ability to simulate future Arctic soil processes through to 2100."

86: what does 'not appropriate' mean in this context? Why is it not appropriate?

To better clarify this, we revise this text as follows:

"Further, the CLM5.0 default soil moisture threshold for decomposition ($\Psi_{min}$ = -2) is too high to permit sub-zero degree soil respiration and this has been identified as a limitation in winter simulations (Tao et al., 2021, Dutch et al., 2023). Similarly, CLM5.0 default settings of Q10 (1.5) and Q10ch4 (1.3) which dictate respiratory responses to changes in temperature are too low for Arctic tundra environments (Dutch et al., 2023, Müller et al., 2015). These parameters $\Psi_{min}$, Q10 and Q10ch4, alongside $K_{eff}$, require adjustment to realistically simulate soil respiration (SR) and methane flux (FCH4) under cold season conditions. Implementing the Sturm et al. (1997) parameterisation reduces the cold bias in simulated soil temperatures by two-thirds. Further, while default CLM5.0 produces negligible winter NEE, combining the Sturm et al. (1997) parameterisation with a mid-range value for $\Psi_{min}$ (-20) produces winter NEE values consistent with field observations (Dutch et al.

**2023, Figure 5). Our study builds on the findings of Dutch et al. (2023) by investigating the influence of the Sturm et al. (1997) parameterisation on future projections of soil temperature and associated carbon emissions."**

161: note the spread of Q values…these are related to the chemical composition of the respired material and can change quite a lot, especially with respect to more labile carbon inputs that are easier to decompose.

**We agree that Q10 is an important and complex variable in controlling soil decomposition which is greatly influenced by the composition of respired material. To capture this variability we implement a wide range of Q10 and Q10ch4 values ranging from 1.3 to 7.5 to reflect temperature sensitivities of both labile and recalcitrant carbon stocks (Fierer et al., 2005, Yan et al., 2017).**

**To reflect this, we add the following to the Methodology:**

**"We implemented a broad range of Q10 and q10ch4 (1.5 – 7.5) to capture variability in the temperature sensitivity of soil respiration associated with differences in carbon pool lability (Fierer et al., 2005, Yan et al., 2017)."**

222: note reference formatting error

**This error has been corrected in the revised manuscript.**

The analysis in Figure 3 is interesting but for the particular site is there a measured data record to compare against, especially because the 2016-2046 averaging period includes the present day? It's critical to understand how well modeled values match measurements to help instill confidence in the future projections.

**We agree that grounding projections in observational data is essential. As mentioned in response to a previous comment, Dutch et al. (2023) conducted a thorough comparison between CLM5.0 simulations and field measurements from TVC including snow depth, soil temperature and Net Ecosystem Exchange (NEE) measurements. Their study shows that incorporating the Sturm et al. (1997) $K_{eff}$ parameterisation improves model performance in winter, alleviating soil temperature biases.**

**References**

Callaghan, T. V., Johansson, M., Brown, R. D., Groisman, P. Y., Labba, N., Radionov, V., Bradley, R. S., Blangy, S., Bulygina, O. N., Christensen, T. R., Colman, J. E., Essery, R. L. H., Forbes, B. C., Forchhammer, M. C., Golubev, V. N., Honrath, R. E., Juday, G. P., Meshcherskaya, A. V., Phoenix, G. K., Pomeroy, J., Rautio, A., Robinson, D. A., Schmidt, N. M., Serreze, M. C., Shevchenko, V. P., Shiklomanov, A. I., Shmakin, A. B., Sköld, P., Sturm, M., Woo, M.-K. & Wood, E. F. 2011. Multiple effects of changes in arctic snow cover. *AMBIO,* 40**,** 32-45.

Dutch, V. R., Rutter, N., Wake, L., Sonnentag, O., Hould Gosselin, G., Sandells, M., Derksen, C., Walker, B., Meyer, G., Essery, R., Kelly, R., Marsh, P., Boike, J. & Detto, M. 2023. Simulating net ecosystem exchange under seasonal snow cover at an arctic tundra site. Copernicus GmbH.

Fahnestock, J. T., Jones, M. H. & Welker, J. M. 1999. Wintertime co2 efflux from arctic soils: Implications for annual carbon budgets. *Global Biogeochemical Cycles,* 13**,** 775-779.

Fierer, N., Craine, J. M., Mclauchlan, K. & Schimel, J. P. 2005. Litter quality and the temperature sensitivity of decomposition. *Ecology,* 86**,** 320-326.

Jordan, R. E. 1991. A one-dimensional temperature model for a snow cover: Technical documentation for sntherm. 89.

Müller, J., Paudel, R., Shoemaker, C., Woodbury, J., Wang, Y. & Mahowald, N. 2015. Ch 4 parameter estimation in clm4. 5bgc using surrogate global optimization. *Geoscientific Model Development,* 8**,** 3285-3310.

Pongracz, A., Wårlind, D., Miller, P. A. & Parmentier, F.-J. W. 2021. Model simulations of arctic biogeochemistry and permafrost extent are highly sensitive to the implemented snow scheme in lpj-guess. *Biogeosciences,* 18**,** 5767-5787.

Schimel, J. P., Bilbrough, C. & Welker, J. M. 2004. Increased snow depth affects microbial activity and nitrogen mineralization in two arctic tundra communities. *Soil Biology and Biochemistry,* 36**,** 217-227.

Schimel, J. P., Fahnestock, J., Michaelson, G., Mikan, C., Ping, C.-L., Romanovsky, V. E. & Welker, J. 2006. Cold-season production of co2 in arctic soils: Can laboratory and field estimates be reconciled through a simple modeling approach? *Arctic, Antarctic, and Alpine Research,* 38**,** 249-256.

Sturm, M., Holmgren, J., König, M. & Morris, K. 1997. The thermal conductivity of seasonal snow. *Journal of Glaciology,* 43**,** 26-41.

Tao, J., Zhu, Q., Riley, W. J. & Neumann, R. B. 2021. Improved elmv1-eca simulations of zero-curtain periods and cold-season ch 4 and co 2 emissions at alaskan arctic tundra sites. *The Cryosphere,* 15**,** 5281-5307.

Welker, J., Fahnestock, J. & Jones, M. 2000. Annual co2 flux in dry and moist arctic tundra: Field responses to increases in summer temperatures and winter snow depth. *Climatic Change,* 44**,** 139-150.

Yan, D., Li, J., Pei, J., Cui, J., Nie, M. & Fang, C. 2017. The temperature sensitivity of soil organic carbon decomposition is greater in subsoil than in topsoil during laboratory incubation. *Scientific reports,* 7**,** 5181.

---

## Author Response (AR2)

**Author Response for "Snow thermal conductivity controls future winter carbon emissions in shrub-tundra", Rutherford *et al*.**

The authors would like to thank the editor and reviewer for taking the time to read and review the revised manuscript and for the suggested improvements. The reviewer comments are included here in **black** and our initial responses are in green. Where we refer to line numbers in our response, this relates to the original submitted document.

**Reply to reviewer 2 comments:**

The revised manuscript is much improved but I'm still a bit surprised at the choice of rcp 8.5 given that it's been widely derided as unrealistic and I feel that the manuscript in many places reads like these dynamics are likely to happen rather than just model realizations. Or perhaps this is just my perception.

We much appreciate the reviewer's opinion that the manuscript is much improved. In terms of inference in the manuscript that we suggest the RCP 8.5 simulation will happen, we found it a little surprising that this was not an issue raised in the initial review. We have tried hard to illustrate uncertainties in all projections by inclusion of both RCP 4.5 and 8.5 throughout the manuscript. While RCP 8.5 may not be attained, the future is likely to be closer to RCP 8.5 than 4.5 towards the end of the century. On re-reading the manuscript to look for unintended messages that model projections (particularly RCP 8.5) were actual realizations, we struggled to find any. Instead, the thrust of our message that $CO_2$/$CH_4$ projections from the representation of snow is as large as the climatic variability in the NA-CORDEX datasets remains quantitatively explicit in the description of uncertainty in simulations.

I like how the revised snow physics module introduces numerous improvements, but because CO2 and methane flux are largely dependent on labile inputs from vegetation growth, even during the shoulder and winter seasons, the lack of focus on how the models treat these processes, especially with respect to future changes, diminishes confidence in the results. This is not to say that the model is in any way bad, it certainly represents an improvement, but the focal areas should be more on model improvements rather than overconfident statements like 'This alignment between observations and simulations provides confidence in the model's ability to simulate future Arctic soil processes through to 2100' on line 318. Keeping the message on the importance of snow parameterizations would improve the flow and help inspire further model improvements.

Correct model process representation of microbial processing of labile carbon is indeed important. And how in the shoulder season this impacts temporally aggregated fluxes to and from the atmosphere would benefit from greater consideration. However, as the reviewer states, our improvements to snow physics are a very welcome development and push forward our capacity to provide plausible simulations of future environmental change. This will only benefit future, more explicit model representation of microbial processing of labile carbon.

We in no way wished to suggest our model improvements for carbon fluxes under tundra snow were an end game, so we welcome the reviewer's opinion that this may have been an unintended consequence of our revised text. Consequently, we have 'softened' the tone of the statement designed to reaffirm to the reader that the Q10 and $\psi_{min}$ parameter values of Dutch et al. 2024 are much more appropriate for Arctic tundra soils than CLM5.0 default values. Consequently, we have changed "This alignment between observations and simulations provides confidence in the model's ability to simulate future Arctic soil processes through to 2100" to "This alignment between observations and simulations increases confidence in the model's ability to plausibly simulate future Arctic soil processes through to 2100"

87: units? Should be MPa

These missing units have now been added.

104 and elsewhere: use a non-breaking space between numbers and units like between all the degree Celsius symbols. Line 235 for example does this more or less correctly.

Spaces between numbers and units are now non-breaking spaces.

The discussion could be reorganized a bit for readability, at the moment it is 3 massive paragraphs.

The discussion is edited and restructured to be more concise and to contain 4 paragraphs for readability, as follows:

1.  We remove the following text to emphasise that this paragraph discusses the future:

"Winter snow cover has a significant influence on the ground thermal regime (Zhang, 2005) and it's representation within ESMs is critical for simulated soil temperatures. The more realistic Keff parameterisation in the CORDEX-Sturm simulations results in reduced soil temperature biases in CLM5.0 (Dutch et al., 2023) and elevates simulated winter soil temperatures to between -10 and 0 °C . Within this range, respiration rates begin to increase rapidly (Natali et al., 2019). Cumulative winter $CO_2$ simulations (Figure 5) under present day conditions are in line with contemporary (2016-2019) NEE simulations, generated using in-situ meteorological data at TVC (Dutch et al., 2024) for both Jordan (~15-35 g$CO_2$ m-2) and Sturm (~25-55 g$CO_2$ m-2) Keff parameterisations."

And add:

 "Soil temperatures are linked to changes in the atmosphere by the thermal conductivity of the snow layer"

2.  We remove the following text:

"Natali et al. (2019) estimated that winter $CO_2$ emissions would increase by 41% by 2100 (from 'present', i.e. 2003-2017 conditions; Natali et al. (2019); this study 2016-2046) under RCP 8.5 whereas CLM5.0 simulations of $CO_2$ emissions under CORDEX-Sturm more than triple under RCP 8.5 which demonstrates the impact of snow representation on simulated soil carbon emissions. Under RCP 8.5 the magnitude of the influence of CORDEX-Sturm on winter carbon fluxes is comparable to the uncertainty in the future climate, reinforcing the importance of snow representation in future projections of Arctic carbon fluxes."

And rework this paragraph to be more concise:

"By comparison, Natali et al. (2019) projected a 41% increase in $CO_2$ emissions by 2100 compared to present. Additionally, under RCP 8.5, the magnitude of the influence of CORDEX-Sturm on winter carbon fluxes is comparable to the uncertainty associated with future climate projections, and snow-related processes - snow cover extent and duration (Natali et al. 2019) and snow thermal conductivity (this study) - emerge as common critical drivers of this uncertainty."

3. **We remove the following text:**

Soil moisture and temperature are critical controls of soil carbon emissions and adjustments to $\Psi_{min}$, $Q_{10}$ and $Q_{10ch4}$ bring CLM5.0 simulations into closer alignment with field measurements (Dutch et al., 2023)."

"Further, the difference in $CO_2$ output between RCP 4.5 and 8.5 shows the possible impacts of climate mitigation efforts on future Arctic winter carbon emissions (Figure 6)."

And more the following text to later in the paragraph:

"Future $CO_2$ and $CH_4$ emissions show greater seasonal variability under CORDEX-Sturm compared with CORDEX-Jordan, particularly under RCP 8.5, which suggests that soil moisture and thermal dynamics are more sensitive to snow cover in the CORDEX-Sturm configuration (Figure E1)."

4. **Finally we remove the following text to better focus on our site specifically (TVC), to be more concise and to keep the focus on future emissions:**

In the recent past (1950 to 2017), both zero-curtain and cold season $CO_2$ emissions have increased, for one site of 0.17 and 0.36 gC m-2 yr-1 at Atqasuk, Alaska (Tao et al., 2021), and CLM5.0 simulations suggest that this is set to continue to 2100. Such increases in soil temperature and zero-curtain duration demonstrates both the influence of snow on soil temperatures at depth and the risks of climate warming on permafrost degradation and possible mobilisation of legacy carbon from Arctic soils.